# Dense shelf-water and associated sediment transport in the Cap de Creus Canyon and adjacent shelf under mild winter regimes: insights from the 2021-2022 winter

- Marta Arjona-Camas <sup>1,2,\*</sup>, Xavier Durrieu de Madron<sup>2</sup>, François Bourrin<sup>2</sup>, Helena Fos<sup>1</sup>, Anna Sanchez-
- Vidal<sup>1</sup>, David Amblas<sup>1</sup>

- ¹GRC Geociències Marines, Departament de Dinàmica de la Terra i de l'Oceà, Universitat de Barcelona, Barcelona, Spain.
- <sup>2</sup>CEFREM, UMR-5110 CNRS-Université de Perpignan Via Domitia, Perpignan, France.
- Correspondence to: Marta Arjona-Camas (<u>marjona@ub.edu</u>)

Abstract. Dense shelf water cascading (DSWC) is a key oceanographic process in transferring energy and matter from continental shelves to deep ocean areas, yet its dynamics under mild winter regimes remain poorly characterized. This study investigates shelf-slope transports of dense shelf waters and suspended particulate matter in the Cap de Creus Canyon (northwestern Mediterranean) during the mild winter of 2021-2022. Observations from the FARDWO-CCC1 multiplatform cruise in March 2022 revealed the presence of dense shelf waters on the continental shelf, which were transported to the canyon head. These cold, dense, and turbid waters, rich in dissolved oxygen and chlorophyll-a, downwelled along the canyon's southern flank to ~390 m depth. Estimated water and suspended sediment transports during this event were 0.3 Sv and 10<sup>5</sup> metric tons, respectively, mainly confined to the upper canyon. Mid-canyon transports were lower (0.05 Sv and 10<sup>4</sup>, respectively), suggesting that during mild winters, dense waters either remain on the shelf, flow along the coast, or cascade only to the upper section of the Cap de Creus Canyon. The Mediterranean Sea Physics reanalysis data indicate that the cascading season lasted from late-January to mid-March 2022, with several shallow cascading pulses, peaking in mid-March during an easterly/southeasterly storm. Our results show that significant export of dense shelf water (260 km³) and suspended sediment can occur in the Cap de Creus Canyon under more moderate atmospheric forcings and shallow cascading. Nevertheless, mild DSWC events likely make a significant contribution to the Western Intermediate Water (WIW) body in the canyon.

**Keywords:** dense shelf water cascading; Western Intermediate Water; mild winter; sediment transport; reanalysis; Gulf of Lion; Mediterranean Sea.

#### 1. Introduction

Continental margins are dynamic transition zones that connect the terrestrial and ocean systems, and play a key role in the transfer and redistribution of particulate material (Nittrouer et al., 2009; Levin and Sibuet, 2012). These regions often receive substantial inputs of sediment, especially from rivers and atmospheric deposition (Blair and Aller, 2012; Liu et al., 2016; Kwon et al., 2021). While much of this material accumulates on the continental shelf, various hydrodynamic processes, such as storm waves, river floods, or ocean currents, can resuspend and transport sediments towards the continental slope and deeper areas of the ocean (Walsh and Nittrouer, 2009; Puig et al., 2014).

One particularly efficient mechanism of shelf-to-slope exchanges is dense shelf water cascading (DSWC). DSWC is a seasonal oceanographic phenomenon that occurs in marine regions worldwide. DSWC starts when surface waters over the continental shelf become denser than surrounding waters by cooling, evaporation, or sea-ice formation with brine rejection (e.g., Ivanov et al., 2004; Durrieu de Madron et al., 2005; Canals et al., 2006; Amblas and Dowdeswell, 2018; Mahjabin et al., 2019, 2020; Gales et al., 2021). Dense shelf waters then sink, generating near-bottom gravity flows that move downslope, frequently funneled through submarine canyons (Allen and Durrieu de Madron, 2009). This process contributes to the ventilation of the deep ocean, and plays an important role in the global thermohaline circulation. Additionally, it modifies the

seabed along its path by eroding and depositing sediments, and facilitates the transfer of energy and matter (including sediment particles, organic carbon, pollutants, and litter) from shallow to deep waters (Canals et al., 2006; Tubau et al., 2013; Puig, 2017; Amblas and Dowdeswell, 2018).

61

The Gulf of Lion (GoL), located in the northwestern Mediterranean, is a very interesting region to study DSWC and shelf-slope exchanges. This region receives important inputs from riverine discharge and atmospheric deposition, and is subject to intense physical forcing (Durrieu de Madron et al., 2008). Episodic events, such as storms or DSWC, efficiently contribute to the export of shelf waters and sediments towards the open sea and deeper parts of the basin (Millot, 1990; Canals et al., 2006; Palanques et al., 2006, 2008; Ogston et al., 2008). The formation of dense waters in the GoL essentially occurs in winter, triggered by persistent (> 30 consecutive days) northerly and north-westerly winds (locally named Tramontane and Mistral) that induce surface heat loss, evaporation, vertical mixing, and the subsequent densification of shelf waters (Durrieu de Madron et al., 2005, 2008; Canals et al., 2006). These dense waters sink until reaching their equilibrium depth and propagate along the western coast of the GoL (Fig. 1). The combined effect of the prevailing cyclonic coastal circulation and the narrowing of the continental shelf near the Cap de Creus Peninsula (Fig. 1), concentrates most of this dense water at the southwestern end of the GoL, where it is mainly exported through the Lacaze-Duthiers and Cap de Creus canyons (Ulses et al., 2008b; Palanques et al., 2008). In these canyons, dense waters overspill the shelf edge and cascade down the slope (Béthoux et al., 2002; Durrieu de Madron et al., 2005), contributing to the ventilation of intermediate and deep layers. As they descend, these overflows reshape the seabed by eroding and depositing sediments, and promote the downslope transport of organic matter accumulated on the shelf. Ultimately, DSWC in the GoL has been observed to impact biogeochemical cycles and the functioning of deepsea ecosystems (Bourrin et al., 2006, 2008; Heussner et al., 2006; Sanchez-Vidal et al., 2008).

**Figure 1.** (a) Bathymetric map of the GoL showing the rivers discharging into the gulf and the incised submarine canyons. Black arrows depict Tramontane and Mistral winds. The grey arrow indicates the direction of the Northern Current over the study area. (b) Bathymetric map of the southwestern part of the Gulf of Lion showing the location of Lacaze-Duthiers Canyon

(LDC) and Cap de Creus Canyon (CCC). Orange triangles indicate the location of buoys (POEM, Banyuls, and SOLA). The location of the re-analysis grid-points at the shelf break (RA-SB) and within the CCC (RA-C) are indicated with dark blue dots. Green dots represent the location of CTD stations carried out during the FARDWO-CCC1 cruise. Red stars mark the location of two instrumented mooring lines deployed at ~1000 m depth at the axis of LDC (LDC-1000) and CCC (CCC-1000). The solid yellow line indicates the glider section on the continental shelf adjacent to the CCC. Dashed lines correspond to the virtual transects defined using reanalysis data: dark blue and light blue lines indicate the location of the upper and mid-canyon transects, respectively, used to estimate dense water transport during winter 2021-2022; the fuchsia dashed line marks the central canyon transect used to determine the temporal evolution of cascading events over the past 26 years (1997-2022).

Due to its relative ease of access, the GoL is particularly suitable for investigating DSWC. However, because of its intermittent nature, DSWC remains difficult to observe directly and its contribution to shelf-deep ocean exchanges is challenging to quantify. Several studies in the GoL have focused on the impact of intense DSWC (IDSWC) on shelf-slope exchanges. These particularly extreme events were monitored in the winters of 1998-1999 (Heussner et al., 2006), 2004-2005 (Canals et al., 2006; Ogston et al., 2008; Puig et al., 2008), 2005-2006 (Pasqual et al., 2010; Sanchez-Vidal et al., 2008), and 2011-2012 (Durrieu de Madron et al., 2013; Palanques and Puig, 2018), with instrumented mooring lines deployed at ~1000 m depth in Lacaze-Duthiers and Cap de Creus Canyons. One of the most studied IDSWC events occurred in the severe winter of 2004-2005. Over 40 days, more than two thirds (750 km³) of GoL's shelf waters cascaded through the Cap de Creus Canyon (Canals et al., 2006; Ulses et al., 2008b), maintaining cold temperatures (11-12.7 °C), high down-canyon velocities (> 0.8 m·s<sup>-1</sup>) and high suspended sediment concentrations (30-40 mg·L<sup>-1</sup>). This DSWC event also transported ~0.6·10<sup>6</sup> tons of organic carbon downcanyon, a mass comparable to the mean annual solid transport of all rivers discharging into the GoL (Canals et al., 2006; Tesi et al., 2010).

Although IDSWC events have drawn particular attention due to their significant impacts, mild DSWC (MDSWC) events are in fact the most frequent since the set of the observational era in the GoL, and they are expected to become more prevalent under the climate change scenario (Herrmann et al., 2008; Durrieu de Madron et al., 2023). These events typically occur during mild winters, characterized by quieter atmospheric conditions and lower heat losses (Martín et al., 2013; Rumín-Caparrós et al., 2013; Mikolajczak et al., 2020). During such winters, the cold continental winds that drive dense water formation are weaker and less persistent, typically blowing for fewer than 30 days and rarely exceeding speeds of 10 m·s<sup>-1</sup>. In contrast, easterly and southeasterly winds become unusually more frequent and sustained, sometimes blowing for more than three consecutive days (Mikolajczak et al., 2020). At the same time, river discharges, particularly from the Rhône River, supply freshwater to the GoL's shelf, further limiting the densification of shelf waters (Ulses et al., 2008a). As a result, most of the dense shelf waters are mixed with lighter ambient waters on the shelf before reaching the slope, and only a small fraction is exported along the coast or to the upper slope, where it contributes to the body of Western Intermediate Water (Duffau-Juliand et al., 2004; Herrmann et al., 2008). Ulses et al. (2008a) estimated a total export of dense shelf waters of 75 km<sup>3</sup> in the mild winter of 2003-2004, an order of magnitude smaller than during IDSWC events. Similarly, during the winter of 2010-2011, only brief and shallow cascading was observed in the Cap de Creus Canyon down to 350 m depth (Martín et al., 2013; Rumín-Caparrós et al., 2013). More recently, modeling studies have expanded our understanding on the interannual variability of dense water export under different atmospheric scenarios, and emphasized the role of moderate winters in driving MDSWC events in the GoL (Mikolajczak et al., 2020).

Nonetheless, most existing studies have relied on numerical simulations and time series from fixed mooring stations, which offer limited spatial resolution and lack direct observations of shelf-slope exchanges. To date, there has been no comprehensive observational characterization of dense water and sediment transport from the shelf to the slope in the Cap de Creus Canyon under moderate winter conditions during MDSWC events. To address this gap, the present study documents

and characterizes MDSWC in the Cap de Creus during the winter of 2021-2022. We combine hydrographic and velocity measurements collected within the canyon and on the adjacent shelf to investigate the shelf-to-slope transport of dense waters and suspended sediment, along with reanalysis data to determine the temporal extent of this event and place it in the context of cascading events observed in the Cap de Creus Canyon over the past 26 years. By focusing on a mild winter scenario, we aim to provide insights into MDSWC events and discuss the implications for WIW formation and deep-sea ecosystems.

# 2. Study area

#### 2.1. General setting

The GoL is a micro-tidal river-dominated continental margin that extends from the Cap Croisette Peninsula, in the northern part of the gulf, to the Cap de Creus Peninsula at its southwestern limit (Fig. 1a). It is characterized by a wide continental shelf (up to 70 km) and it is incised by a dense network of submarine canyons (Lastras et al., 2007).

The GoL is impacted by strong continental winds from the north and northwest (Mistral and Tramontane, respectively), which favor dense water formation and cascading events in winter (Durrieu de Madron et al., 2013) and coastal upwellings in summer (Odic et al., 2022). Humid easterly and southeasterly winds, which blow less frequently, occur primarily from autumn to spring, and can produce large swells and high waves over the continental shelf (Ferré et al., 2005; Leredde et al., 2007; Petrenko et al., 2008; Guizien, 2009). They do not produce a densification of surface waters but lead to an intense cyclonic circulation on the GoL's shelf and to a strong export of shelf waters at the southwestern exit of the GoL (Ulses et al., 2008a; Mikolajczak et al., 2020).

The ocean circulation of the GoL is also influenced by the freshwater inputs from the Rhône River, one of the largest Mediterranean Rivers, and a series of smaller rivers with typical flash-flooding regimes (Ludwig et al., 2009). The Rhône River supplies an annual river discharge of 1700 m³·s·¹ and an average of 8.4 Mt·y¹ of suspended particulate matter (Sadaoui et al., 2016; Poulier et al., 2019). The smaller rivers (from south to north the Tech, Têt, Agly, Aude, Orb, Hérault, Lez, and Vidourle) account for slightly 5% of the total inputs of particulate matter (~0.5 Mt·y⁻¹) to the GoL (Serrat et al., 2001; Bourrin et al., 2006). Most of the sediment delivered by these rivers is temporarily stored near their mouths in deltas and prodeltas (Drexler and Nittrouer, 2008), and afterwards it is remobilized and redistributed along the margin by the action of storms and wind-driven along-shelf currents (Estournel et al., 2023). This westerly current carries particulate material from the shelf towards the southwestern end of the GoL, where the continental shelf rapidly narrows and the Cap de Creus Peninsula constraints the circulation, intensifying the water flow and increasing the particle concentration (Durrieu de Madron et al., 1990; Canals et al., 2006).

The Cap de Creus Canyon represents the limit between the GoL and the Catalan margin (Fig. 1b). The canyon head is located only 4 km from the coast and incises the shelf edge at 110-130 m depth (Lastras et al., 2007). Due to its proximity to the coast and the preferential direction of coastal currents, this canyon has been identified as the main pathway for the transfer of water and sediments from the shelf to the slope and deep margin (Canals et al., 2006; Palanques et al., 2006, 2012).

# 2.2. Hydrography

The surface layers over the GoL shelf stratify between late spring and autumn (Millot, 1990). In winter, surface cooling and wind-driven mixing weaken the stratification, which leads to a vertically homogeneous water column over the continental shelf (Durrieu de Madron and Panouse, 1996). Offshore, the GoL is characterized by several water masses. The Atlantic Water (AW) fills the upper 250 m and flows into the Mediterranean Sea through the Strait of Gibraltar. During its transit to the GoL, its hydrographic properties undergo chemical and physical modifications, becoming the "old" Atlantic Water (oAW) with T > 13 °C and S = 38.0-38.2 (Millot, 1999). Below, the Eastern Intermediate Water (EIW) occupies the water column between 250 and 850 m depth. It is formed during winter in the Levantine Basin, in the Eastern Mediterranean Sea, by intermediate convection (Font, 1987; Millot, 1999; Taillandier et al., 2022; Schroeder et al., 2024). The Western

Mediterranean Deep Water (WMDW) occupies the deepest bathymetric levels and is formed in winter at interannual scales in the GoL by open-ocean deep convection. It typically shows T ~ 13 °C and S = 38.43-38.47 (Marshall and Schott, 1999; Somot et al., 2016). During winter, northerly and north-westerly winds cause the formation of dense shelf waters (DSW) over the GoL's shelf (T < 13 °C and S < 38.4) (Houpert et al., 2016; Testor et al., 2018). During mild winters, these dense waters do not gain enough density ( $\sigma$  < 29.05 kg·m<sup>-3</sup>) to sink into the deep basin, and contribute to the Western Intermediate Water (WIW) (T = 12.6-13.0 °C and S = 38.1-38.3) body found at upper slope depths (~380-400 m) (Dufau-Julliand et al., 2004; Durrieu de Madron et al., 2005; Juza et al., 2013). The formation of WIW is an important process in the Mediterranean Thermohaline Circulation (MTHC), as it contributes to the ventilation of intermediate layers and plays a role in preconditioning the region for deeper convection events (Juza et al., 2019). During extreme winters, the potential density of DSW exceeds that of the EIW ( $\sigma$  = 29.05-29.10 kg·m<sup>-3</sup>) and even surpasses the density of the WMDW ( $\sigma$  = 29.10–29.16 kg/m<sup>3</sup>), enabling DSWC to reach the deep basin around 2000-2500 m depth. This process contributes to the ventilation of the deep waters and to the final characteristics of the WMDW (Durrieu de Madron, 2013; Palanques and Puig, 2018).

# 3. Materials and methods

# 3.1. Field data

# 3.1.1. Hydrographic transects and current vertical profiles

During the FARDWO-CCC1 Cruise, onboard the R/V *García del Cid*, two hydrographic transects were conducted on March 5-6, 2022, across the Cap de Creus Canyon, with stations spaced every 1.5 km (Fig. 1b). The T1 transect covered the upper section of the canyon, while the T2 transect covered the mid-canyon section. The hydrographic data were collected using a SeaBird 9 CTD probe coupled with a SeaPoint Fluorometer and Turbidity Sensor (700 nm), an SBE43 dissolved oxygen sensor, and a WetLabs C-Star Transmissometer (650 nm). These sensors were mounted on a rosette system with twelve 12 L Niskin bottles, which collected water samples from various depths. The CTD-Rosette system was hauled through the water column, allowing the comparison of sensor outputs in productive surface waters with high fluorescence, in clear midwaters, and in intermediate and bottom nepheloid layers loaded with fine suspended particulate matter. The hydrographic profiles obtained from the CTD stations were interpolated onto a regular grid using the isopycnic gridding method. This method is an advance gridding procedure that aligns property contours along lines of constant potential density (isopycnals) rather than constant depth, which improves the representation of water mass structure and reduces artificial smoothing across density gradients (Schlitzer, 2022).

Velocity profiles were simultaneously measured using two Lowered Acoustic Doppler Current Profiler (LADCPs) Teledyne 300 kHz narrowband units mounted on the CTD frame. One unit faced upwards and the other downwards to maximize the total range of velocity observations. Each instrument was configured with 20 bins, each 8 m in length, which allowed for a measurement range of ~160 m (Thurnherr, 2010). The raw LADCP data were processed using the LDEO Software version IX by applying the least-squared method, using the navigation and the vessel-mounted (VM) ADCP data as constraints (Visbeck, 2002).

# 3.1.2. Vessel-mounted (VM) ADCP data

The R/V *Garcia del Cid* was equipped with a vessel-mounted (VM) 75 kHz ADCP (Ocean Surveyor, RD Instruments) that recorded current velocities along the hydrographic sections. The VM-ADCP was configured to record current velocities at 2-min averages, while sampling depths were set to 8-m vertical bins. The VM-ADCP data were processed using the CASCADE V7.2 software (Le Bot et al., 2011). Absolute current velocities resulted from removing the vessel speed, derived from a differential global positioning system. The measurement range of the instrument was from 20 m to 720 m depth. Given the micro-tidal regime of the Mediterranean Sea (tidal range ~ 0.5 m), tides are considered of very low magnitude in the GoL (Davis and Hayes, 1984; Dipper, 2022). Nevertheless, a barotropic tide correction was applied to the VM-ADCP data using

the OTIS-TPX8 Tide model (<a href="https://www.tpxo.net/global">https://www.tpxo.net/global</a>) within the cascade V7.2 software, which provided tide-corrected current velocities. VM-ADCP data were then screened to align with the timeframe of the different hydrographic sections. However, the VM-ADCP's range could not fully cover the water column due to the interaction of secondary acoustical lobes with the seabed. To address this limitation, we integrated VM-ADCP data with LADCP measurements recorded at each station. This integration involved aligning time stamps, matching coordinate systems, and merging and interpolating datasets.

Finally, ADCP velocities, decomposed into E-W and N-S components, were rotated 15° counterclockwise to align with the local canyon axis orientation. This rotation provided the across- and along-canyon velocity components for interpretation. For the across-canyon component, eastward velocities have been defined as positive and westward velocities as negative. For the along-canyon component, up-canyon velocities are positive, while down-canyon velocities are negative.

#### 3.1.3. Glider observations

The hydrodynamics of dense shelf waters at the continental shelf adjacent to the Cap de Creus Canyon were monitored by a SeaExplorer underwater glider (ALSEAMAR-ALCEN). The glider carried out one along-shelf (i.e., north-south) section and navigated 10 km for 25 hours, starting on March 5, 2022 (Fig. 1b). It conducted a total of 28 yos (down/up casts). It followed a sawtooth-shaped trajectory through the water column, descending typically to 2 m above the bottom (between 83 and 92 m depth) and ascending to 0-1 m of distance from the surface. The glider was equipped with a pumped SeaBird GPCTD, coupled with a dissolved oxygen sensor (SBE43F), that acquired temperature, conductivity, and pressure data at a sampling rate of 4s. Additionally, the glider had a SeaBird Triplet (WetLabs FLBBCD sensor), which measured proxies of phytoplankton abundance (by measuring the fluorescence of chlorophyll-a at 470/695 nm), and total particle concentration (by measuring optical backscattering at 700 nm). Finally, the glider integrated a downward-looking AD2CP (Acoustic Doppler Dual Current Profiler) that measured relative water column velocities and the glider speed. The AD2CP was configured with 15 vertically stacked cells of 2 m resolution, and a sampling frequency of 5 s. Raw velocity measurements were processed using the shear method to derive absolute current velocities (Fischer and Visbeck, 1993; Thurnherr et al., 2015; Homrani et al., 2025). We used bottom tracking and GPS coordinates as constraints to reference each velocity profile.

The hydrographic profiles obtained from the glider were similarly interpolated using the isopycnic gridding method as applied to the CTD data (Schlitzer, 2022).

#### 3.2. Ancillary data

#### 3.2.1. Environmental forcings and shelf observational data

Heat flux data were used to evaluate the interaction between the atmosphere and the ocean surface. Data were retrieved from the European Centre for Middle-range Forecast (ECMWF) ERA5 reanalysis by Copernicus Climate Change Service. This global reanalysis product provides hourly estimates of a large number of atmospheric variables since 1940, with a horizontal resolution of 0.25 ° (Hersbach et al., 2023). For this study, the sea-atmosphere interactions were assessed from 42.6 °N to 43.4 °N and from 3 °E to 4.5 °E, approximately covering the entire GoL's continental shelf, from October 2021 to July 2022.

Significant wave height (Hs) data were retrieved from the Banyuls buoy (06601) located at 42.49 °N and 3.17 °E (Fig. 1b), through the French CANDHIS database (<a href="https://candhis.cerema.fr">https://candhis.cerema.fr</a>). These measurements were recorded every 30 min using a directional wave buoy (Datawell DWR MkIII 70). Wind speed and direction data were obtained from the POEM buoy (<a href="https://www.seanoe.org/data/00777/88936/">https://www.seanoe.org/data/00777/88936/</a>; Bourrin et al., 2022), which records hourly observations using a surface meteorological sensor and is also equipped with a multiparametric CTD probe measuring near-surface temperature and salinity. In addition, the SOLA station, located in the Bay of Banyuls-sur-Mer (Fig. 1) and managed by the SOMLIT network (<a href="https://www.seanoe.org/data/00886/99794/">https://www.seanoe.org/data/00886/99794/</a>; Conan et al., 2024) provided weekly near-bottom (~27 km) time series of temperature, salinity, and pressure. This data allowed to monitor the presence of dense shelf waters over the continental shelf during the winter of 2021-2022. Raw data were processed to remove gaps and outliers prior to analysis.

Water discharge of rivers opening to the GoL was measured by gauging stations located near rivers mouths and provided by Hydro Portail v3.1.4.3 (<a href="https://hydro.eaufrance.fr">https://hydro.eaufrance.fr</a>). For this study, we have considered the river discharge from the Rhône River and the total discharge from coastal rivers as the sum of their individual contribution.

# 3.2.2. Instrumented mooring lines

Current velocity and direction, temperature, and SPM concentration were monitored in two submarine canyons, the Lacaze-Duthiers Canyon (LDC) and the Cap de Creus Canyon (CCC), by means of two instrumented mooring lines (Fig. 1b). The mooring line at Lacaze-Duthiers Canyon, located at about 1000 m depth, has been maintained in the canyon since 1993 as part of the MOOSE program. It is equipped with two current meters (Nortek Aquadopp 2 MHz) with temperature, conductivity, and turbidity sensors, placed at 500 m and 1000 m depth (referred to as LDC-500 and LDC-1000, respectively). The sampling interval for the current meters was set to 60 min. The fixed mooring line at the Cap de Creus Canyon has been maintained in the canyon axis by the University of Barcelona since December 2010. This mooring line, referred to as CCC-1000, is equipped with a current meter (Nortek Aquadopp 2 MHz) with temperature, conductivity, and turbidity sensors placed at 18 m above the bottom (mab). The current meter sampling interval was set to 15 min. For this study, the recording period was considered from December 1, 2021 to April 1, 2022, which encompassed the winter of 2021-2022.

# 3.3. Determination of suspended particulate matter (SPM) concentration from turbidity measurements

To calibrate turbidity measurements obtained in the Cap de Creus Canyon during the FARDWO-CCC1 Cruise, water samples were collected at selected depths with Niskin bottles. For each sample, between 2 and 3 L of water were vacuum filtered into 47 mm 0.4  $\mu$ m pre-weighted Nucleopore filters. The filters were then rinsed with MiliQ water to remove salt and stored at 4 °C. Then, these filters were dried by desiccation and weighted. The weighing difference of the filters divided by the volume filtered of seawater yielded to the SPM concentration, in mg·L<sup>-1</sup>. Finally, the measured SPM concentrations were plotted against in situ turbidity measurements, both from the transmissometer and the optical sensor (Figs. 2a, b). The beam attenuation coefficient (BAC) from the transmissometer showed a stronger correlation with SPM (R<sup>2</sup> = 0.86) compared to FTU from the SeaPoint turbidity sensor (R<sup>2</sup> = 0.81). Therefore, SPM concentrations were predicted using the equation:

$$SPM = 1.84 \cdot BAC + 0.05 (R^2 = 0.85; n = 25)_{(1)}$$

**Figure 2.** Relationship between the weighed SPM mass concentration (mg·L<sup>-1</sup>) and: (a) turbidity records (FTU) from the SeaPoint sensor and (b) BAC (m<sup>-1</sup>) from the WetLabs transmissometer attached to the CTD-rosette system; (c) optical backscatter sensor data (counts) from the WetLabs FLBBCD sensor mounted on the SeaExplorer glider; (d) echo intensity (dB) retrieved from the current meters installed in LDC and CCC instrumented moorings. For each relationship, the regression coefficient (R<sup>2</sup>) is given.

Backscatter data (in counts) from a self-contained FLBBCD optical sensor, identical to the one installed on the SeaExplorer glider, were also correlated to SPM concentration using a laboratory calibration. Firstly, the optical sensor was submerged into a 10-L plastic tank filled with freshwater. Then, sediment was added while stirring the water to distribute the sediment particles evenly. The amount of sediment was gradually increased until the instrument reached a saturation level of approximately 4,000 counts, corresponding to a sediment mass of 447.5 mg. The optical sensor, connected to a computer, displayed and stored the turbidity measurements during the calibration process. Then, water samples were taken at each interval and filtered, obtaining SPM concentration (in mg·L-1). Pairs of counts/SPM data points yielded the following regression line (Fig. 2c):

SPM = 
$$0.01 \cdot \text{counts}$$
 (R<sup>2</sup> =  $0.97$ ; n = 20) (2)

Finally, echo intensity (EI) records from the current meters equipped in LDC and CCC lines were also correlated with SPM concentration, using a linear equation relating the logarithm of SPM to EI (Gartner, 2004). Unlike optical devices, acoustic sensors measure relative particle concentrations based on changes on the backscattered acoustic signal (Fugate and Friedrichs, 2002). Acoustic backscatter, expressed in counts, is proportional to the decibel sound pressure level (Lohrmann, 2001), and depends on the particle size and on the operating frequency of the acoustic sensor (Wilson and Hay, 2015). The equation derived from the regression between backscatter data and direct sampling concentration in the GoL was (Fig. 2d):

$$10 \cdot \log (SPM) = 0.405 \cdot EI - 22.46 (R^2 = 0.94; n = 66)_{(3)}$$

# 3.4. Estimation of dense water and SPM transports from observations

The total transports of dense shelf water and associated SPM were estimated using hydrographic and current measurements collected during the FARDWO-CCC1 cruise and the glider survey conducted in early March 2022 in the Cap de Creus Canyon and its adjacent shelf, respectively.

First, to isolate dense shelf waters, we selected water masses with a potential temperature below 12.9 °C and potential density below 29.05 kg·m $^{-3}$ . At the canyon, these properties already corresponded to the signature of the Western Intermediate Water (WIW). Then, dense water transport (expressed in  $Sv = 1 \cdot 10^6 \text{ m}^3 \cdot \text{s}^{-1}$ ) was calculated by integrating the zonal component of current speed (which aligns with the down-canyon direction) over the area occupied by dense waters. For the T1 and T2 transects, current speed was measured using the LADCP attached to the CTD-rosette system, and transport was calculated in vertical cells of 8 m (the LADCP bin size) and then integrated over the whole transect. For the glider section over the continental shelf, dense water transport was computed using the same approach, although current velocity data were obtained from the glider-mounted AD2CP.

After estimating dense water transport, we computed the associated SPM transport (in metric tons) by multiplying the calculated water transport by the SPM concentration within the dense water layer. SPM concentrations were derived from the SeaPoint sensor integrated into the CTD probe for the canyon transects, and from the WetLabs FLBBCD sensor onboard the glider for the shelf section. Finally, all SPM profiles were resampled to match the vertical resolution of current velocity measurements prior to integration.

#### 3.5. Reanalysis data

 To complement the observational dataset, we used the Mediterranean Sea Physics Reanalysis (hereafter MedSea) from the Copernicus Marine Service to evaluate the spatiotemporal evolution of hydrographic properties and transport patterns of dense shelf waters in the Cap de Creus Canyon. This reanalysis product has a horizontal resolution of 1/24° (~4-5 km) and provides daily estimates of ocean variables, including temperature, salinity, and currents at multiple depths (Escudier et al., 2020, 2021).

First, we used the MedSea data to assess the temporal evolution of hydrographic conditions during winter 2021-2022. Daily time series of temperature and salinity were extracted from two grid points at key locations along the pathway of dense waters: one located at the shelf break (RA-SB: 42.48 °N and 3.29 °E, depth= 250 m), and another within the Cap de Creus Canyon (RA-C: 42.34 °N and 3.37 °E, depth= 350 m) (Fig. 1b).

Then, we calculated the transport of dense shelf waters through the canyon using MedSea data by applying the same approach as for the observational dataset. First, we identified dense waters using the same hydrographic thresholds (T < 12.9 °C and  $\sigma$  < 29.05 kg·m<sup>-3</sup>). Then, we estimated the dense water transport (in Sv) by integrating the zonal velocity component (aligned with the downcanyon direction) across two virtual transects close to the T1 and T2 sections sampled during the FARDWO-CCC1 cruise. This analysis was performed for the winter of 2021-2022 to capture the temporal variability of the transport, and allowed us to extend the analysis beyond the limited period covered by the field data. Finally, to determine the temporal context and interannual variability of dense shelf water transport, we analyzed a 26-year (1997-2022) time series of MedSea reanalysis data across a central transect in the Cap de Creus Canyon (see location in Figure 1). Dense waters were identified by temperature < 12.9 °C and zonal velocity > 0 m·s<sup>-1</sup> (down-canyon velocities). We distinguished IDSWC events by a density threshold > 29.1 kg·m<sup>-3</sup>, whereas MDSWC events were characterized by densities below this threshold. This long-term analysis allowed us to place the 2021–2022 MDSWC event in the context of other cascading episodes previously reported in the Gulf of Lion and to evaluate its relative intensity.

# 4. Results

# 4.1 Meteorological and oceanographic conditions during winter 2021-2022

The time series of net heat fluxes over the GoL's shelf showed negative values from October 2021 to early April 2022, indicating a heat loss from the ocean to the atmosphere (Fig. 3a). The strongest net heat losses during that winter occurred in November 2021 and January 2022, reaching values of about -400 W·m $^{-2}$  (Fig. 3a).

Winter 2021-2022 was characterized by frequent northerly and north-westerly winds (Tramontane and Mistral) with speeds ranging from 5 to 18 m·s<sup>-1</sup> (Fig. 3b). The duration of these wind events varied throughout the season, blowing in short bursts of 1 to 3 days during October and December, and in longer periods of 5 to 7 consecutive days during November and February (Fig. 3b). The highest wind speeds during this period exceeded 20 m·s<sup>-1</sup> and were associated with strong heat losses (Figs. 3b, c). From March to May 2022, the wind pattern alternated between northerly/north-westerly winds and short intervals (2-4 days) of easterly/southeasterly winds (Fig. 3b). Similar to winter, the strongest heat losses occurred in April 2022, and coincided with a relatively long period (5 days) of northerly/north-westerly winds with speeds above 20 m·s<sup>-1</sup> (Figs. 3a, b).

Significant wave height (Hs) ranged between 0.5 and 2.0 m during winter (Fig. 3c). During this period, only one marine storm, defined as sustained Hs > 2 m for more than 6 hours (Mendoza and Jimenez, 2009), was recorded on March 13, 2022. This storm was associated with an easterly/south-easterly wind event with maximum speeds of  $\sim$ 19 m·s<sup>-1</sup>, and generated Hs > 3 m for over 20 hours (Fig. 3c).

The mean water discharge of the Rhône River generally displayed average values of about 1850 m<sup>3</sup>·s<sup>-1</sup> (Fig. 3d; Pont et al., 2002; Provansal et al., 2014). However, the discharge exceeded 2000 m<sup>3</sup>·s<sup>-1</sup> during early October, early November, and mid-December 2021 (Fig. 3d). A peak discharge of over 5000 m<sup>3</sup>·s<sup>-1</sup> occurred in late December, associated with a brief

easterly/south-easterly wind event (Fig. 3c). Coastal river discharges remained relatively low throughout winter (see average daily water discharge values in Bourrin et al., 2006), typically below 200 m<sup>3</sup>·s<sup>-1</sup>, except for a few short episodes when river discharge exceeded 400 m<sup>3</sup>·s<sup>-1</sup> (Fig. 3d). The highest coastal river discharge occurred on March 13, 2022, after a period of sustained easterly/southeasterly winds (Fig. 3c). During this event, the coastal river discharges reached 2265 m<sup>3</sup>·s<sup>-1</sup>, surpassing the water discharge of the Rhône River, which was comparatively lower (884 m<sup>3</sup>·s<sup>-1</sup>) (Fig. 3d).

The near-bottom temperature time series at the continental shelf, measured by the POEM buoy, showed a gradual decline from 20 °C in October 2021 to 13 °C in March 2022 (Fig. 3e). Salinity fluctuated between 37.8 to 38.1 during this period, occasionally dropping below 37.8 in parallel with temperature decreases (Fig. 3f). Near-bottom temperature and salinity time series at the shelf break (250 m depth) and at the Cap de Creus Canyon (350 m depth), as previously noted, were derived from reanalysis data. At the shelf break, temperature remained constant around 14.5 °C, except for an increase to 15.5 °C throughout November and a subsequent decrease below 13 °C from January to late March 2022 (Fig. 3e). Salinity at the shelf break also remained constant (~38.55) throughout winter (Fig. 3f). At the Cap de Creus Canyon, temperature and salinity remained stable around 14.5 °C and 38.8, respectively, during winter. From late January to late March 2022, both variables occasionally dropped in short-lived pulses to 12.9 °C and 38.1, respectively, before returning to baseline values (Figs. 3e, f). The potential density anomaly at the shelf and shelf break followed the seasonal temperature variations, and it only increased during February-March 2022 when it peaked at ~28.9 kg·m<sup>-3</sup> (Fig. 3g). In contrast, density at the Cap de Creus Canyon remained relatively stable, slightly varying between 28.75 and 28.95 kg·m<sup>-3</sup> throughout the study period (Fig. 3g).

**Figure 3.** Time series of (a) net heat fluxes (NHF, W·m<sup>-2</sup>) averaged over the GoL's shelf; (b) wind speed (m·s<sup>-1</sup>) measured at the POEM buoy. Blue and red shaded bars indicate N/NW and S/SE winds, respectively; (c) significant wave height (Hs, m) measured at the Banyuls buoy; (d) river discharge (Q, m<sup>3</sup>·s<sup>-1</sup>) of the Rhône River (dark blue) and coastal river discharges discharging into the GoL (light blue); (e) weekly mean bottom temperature ( $\theta$ , °C), (f) salinity, and (g) potential density ( $\sigma$ , kg·m<sup>-3</sup>) calculated from TS data at the continental shelf (black), the shelf break (orange), and the Cap de Creus Canyon (green). Data at the continental shelf were measured at the POEM buoy. Data at the shelf break and within the Cap de Creus Canyon were extracted from two grid points from the MedSea. All data are displayed for the period between October 2021 and May 2022. The grey shaded vertical bar highlights the FARDWO-CCC1 Cruise, which took place on March 1-7th, 2022. The studied winter period is also indicated in the figure and corresponds to the time of most negative NHF values, indicating the strongest ocean heat loss to the atmosphere and favorable conditions for DSWC events.

#### 4.2 Time series at Lacaze-Duthiers and Cap de Creus canyons during winter 2021-2022

Figure 4 shows the time series of temperature, current speed and direction, and SPM recorded at the mooring stations in Lacaze-Duthiers Canyon (at 500 and 1000 m depth) and the Cap de Creus Canyon (at 1000 m depth) during winter 2021-2022.

**Figure 4.** Time series of (a) potential temperature (°C), (b-d) stick plots of currents and current speed as a shaded grey area (m·s<sup>-1</sup>), and (e) suspended particulate matter concentration (SPM, mg·L<sup>-1</sup>) measured at the mooring stations in Lacaze-Duthiers Canyon (LDC-500 in light blue, and LDC-1000 in dark blue) Cap de Creus Canyon (CCC-1000 in red) during winter 2021-

2022. In panels (b-d), arrows indicate the along-canyon component for each mooring site. The grey shaded vertical bar highlights the FARDWO-CCC1 Cruise, which took place on March 1-7, 2022.

At 500 m depth in Lacaze-Duthiers Canyn (LDC-500), temperature remained relatively constant between 13.5 and 14 °C, while at 1000 m in both Lacaze-Duthiers and Cap de Creus canyons, it fluctuated between 13.1 and 13.4 °C (Fig. 4a). Current speeds ranged from 0.01 to 0.1 m·s<sup>-1</sup> at 500 in Lacaze-Duthiers Canyon and at 1000 m in the Cap de Creus Canyon (Figs. 4b, d), and from 0.1 and 0.2 m·s<sup>-1</sup> at 1000 m in Lacaze-Duthiers Canyon (Fig. 4c). At this depth, current speed increased several times to 0.12 m·s<sup>-1</sup> between January and mid-March 2022, and peaked at 0.2 m·s<sup>-1</sup> in late March (Fig. 4c). A similar, but less intense peak was observed in the Cap de Creus Canyon at 1000 m during the same period (Fig. 4c), coinciding with small temperature drops (Fig. 4a). Current direction at 500 m in Lacaze-Duthiers Canyon was isotropic (Fig. 4b), while at 1000 m in both Lacaze-Duthiers and Cap de Creus canyons, it became anisotropic and preferentially aligned with the orientation of the respective canyon axes (Figs. 4c, d).

The time series of SPM concentration at all mooring sites during winter 2021-2022 showed values ranging between 0.2 and 0.5 mg·L<sup>-1</sup> at all monitored depths (Fig. 4e). The highest SPM concentrations were recorded at 1000 m depth in the Cap de Creus Canyon mid-February to late March 2022, reaching values of  $\sim 1 \text{ mg·L}^{-1}$  (Fig. 4e).

# 4.3. Water column properties at the Cap de Creus Canyon and adjacent shelf in March 2022

# 4.3.1. Hydrographic and biogeochemical properties

The continental shelf adjacent to the Cap de Creus Canyon was monitored on March 5, using a SeaExplorer glider, which monitored the water column from the surface down to 92 m depth. In the northern part of the shelf transect, a water mass with temperatures of 12.9 °C and salinities of 38.2 occupied most of the water column above the 28.95 kg·m<sup>-3</sup> isopycnal (Figs. 5a-c). This water mass exhibited SPM concentrations of 1.5-3.0 mg·L<sup>-1</sup> and dissolved oxygen values of ~232 µmol·kg<sup>-1</sup> (Figs. 6a, c). Near the bottom, a relatively cooler, fresher, and more turbid water mass with higher dissolved oxygen concentrations was observed in the northern part of the section (Figs. 5a-c and 6a-c). This water mass flowed along the shelf below the 28.95 kg·m<sup>-3</sup> isopycnal, gradually becoming slightly warmer, saltier, less turbid, and less oxygenated towards the southern part of the shelf transect (Figs. 5a-c and 6a-c). Fluorescence values generally increased towards the southern part of the shelf transect (Figs. 6b). These hydrographic and biogeochemical properties clearly indicate the presence of dense shelf waters on the continental shelf in the shape of a wedge that thickened towards the Cap de Creus Peninsula (i.e., south) near the bottom (Figs. 5a-c and 6a-c). These results reflect the oceanographic conditions on the continental shelf shortly before the start of the FARDWO-CCC1 cruise.

During the FARDWO-CCC1 cruise, two ship-based CTD transects were conducted across the Cap de Creus Canyon. The T1 transect was carried out across the upper section of the canyon, from the surface down to 625 m depth. The upper water column (< 100 m depth) was characterized by temperatures of 13-13.2 °C, salinities of ~38.25 (Figs. 5d, e), dissolved oxygen concentrations of ~200 µmol·kg<sup>-1</sup>, and relatively low SPM concentrations (0.3 mg·L<sup>-1</sup>) (Figs. 6d, f). Beneath this layer, a colder (12.2-12.7 °C), fresher (S = 38.1-38.2) water mass occupied depths of 100-400 m along the southern flank of the canyon (Fig. 5d, f). This water mass showed relatively high dissolved oxygen values (> 200 µmol·kg<sup>-1</sup>) and increased SPM concentrations (1-1.2 mg·L<sup>-1</sup>) (Figs. 6d-f). It extended ~3 km into the canyon interior. Fluorescence values were generally high (0.5-1.1 µg·L<sup>-1</sup>) in the upper 400 m of the water column (Fig. 6e). Below, a relatively cooler, saltier, less oxygenated, and slightly denser (~29.55 kg·m<sup>-3</sup>) water mass was apparent around 350-400 m depth (Figs. 5d-f). Deeper into the canyon (> 400 m depth), a much warmer (> 13 °C) and saltier (> 38.4) water mass with lower SPM concentrations and fluorescence values was observed (Figs. 5d-f and 6d-f).

Further downcanyon, the T2 transect crossed the middle section of the Cap de Creus Canyon and covered the entire water column down to 850 m depth (Figs. 5g-i and 6g-i). The upper 250 m showed a remarkably homogeneous water mass

throughout the canyon. It was characterized by temperatures of 13.2-13.4 °C, salinities of ~38.2, relatively high dissolved oxygen values (197  $\mu$ mol·kg<sup>-1</sup>), and low SPM concentrations (0.3 mg·L<sup>-1</sup>) (Figs. 5g-i and 6g-i). Below this layer, and down to 380 m depth, a much colder (12.2-12.3 °C), fresher (38.1), highly oxygenated (200-203  $\mu$ mol·kg<sup>-1</sup>) and moderately turbid (0.8-1.0 mg·L<sup>-1</sup>) water mass was observed (Figs. 5g-i and 6g-i). This water mass only occupied the southern flank of the canyon and extended approximately 2.5 km into the interior of the canyon (Figs. 5g-i and 6g-i). Fluorescence values ranged from 0.18 to 0.60  $\mu$ g·L<sup>-1</sup> throughout the canyon from the surface to ~380 m depth above the 29.0 kg·m<sup>-3</sup> isopycnal (Fig. 6h). Finally, between 400 m and 850 m depth, the water column was characterized by temperatures of ~13.1 °C, salinities of 38.4, and relatively lower oxygen concentrations, fluorescence values, and SPM concentration (Figs. 6g-i).

**Figure 5.** Contour plots of potential temperature (°C), salinity, and potential density (σ, kg·m<sup>-3</sup>) at (a-c) the continental shelf (glider transect) and the Cap de Creus Canyon during (d-f) T1 transect and (g-i) T2 transect (FARDWO-CCC1 cruise). Note that for panels (a-c), the section distance goes from south to north, whereas for panels (d-i), the section distance represents a southwest-northeast direction. Note the change in the density scale in panel (c).

**Figure 6.** Contour plots of SPM concentration (mg·L<sup>-1</sup>), fluorescence (μg·L<sup>-1</sup>), and dissolved oxygen concentration (μmol·kg<sup>-1</sup>) at (a-c) the continental shelf (glider transect) and at the Cap de Creus Canyon during (d-f) T1 transect and (g-i) T2 transect (FARDWO-CCC1 cruise). Note that for panels (a-c), the section distance goes from south to north, whereas for panels (d-i) the section distance represents a southwest-northeast direction. Note the change in the dissolved oxygen scale in panel (c).

#### 4.3.2. Currents

At the continental shelf, current velocity measurements indicated a fairly homogeneous current field (Figs. 7a-c). Cross-shelf currents exhibited low velocities (< 0.1 m·s<sup>-1</sup>), predominantly eastward (Fig. 7a), while along-shelf currents were slightly higher (0.1-0.15 m·s<sup>-1</sup>) and predominantly southward (Fig. 7b). The spatial distribution of currents averaged within the dense water plume (defined between the 28.9 and 29.0 kg·m<sup>-3</sup> isopycnals) shows that dense shelf waters flowed south/south-eastward along the continental shelf adjacent to the Cap de Creus Canyon, with speeds generally around 0.15 m·s<sup>-1</sup> (Fig. 7c). However, in the southern part of the section, currents were stronger, exceeding 0.2 m·s<sup>-1</sup> (Fig. 7c).

In the Cap de Creus Canyon, at the upper section (T1 transect), current velocity measurements revealed the highest speeds from the surface to 400 m depth at the southern canyon flank (Figs. 7d, e). The zonal current velocity (E-W) reached up to 0.3 m·s<sup>-1</sup> eastward (Fig. 7d), while the meridional current velocity (N-S) was primarily southward, with velocities between 0.1 and 0.2 m·s<sup>-1</sup> (Fig. 7e). These currents appeared relatively confined within the canyon axis, with a pronounced down-canyon component (Fig. 7f). At the mid-canyon section (T2 transect), the highest current velocities were observed

between 200 and 400 m depth (Figs. 7g, h). The zonal current velocities were more pronounced, and reached values of  $\sim$ 0.15 m·s<sup>-1</sup> eastward (Fig. 7g), while the meridional current velocities were slightly smaller, ranging from 0.1 to 0.12 m·s<sup>-1</sup> southward (Fig. 7h). On the southern flank, currents were generally weaker ( $\sim$ 0.1 m·s<sup>-1</sup>) and predominantly southward, whereas at the northern canyon flank, these were stronger ( $\sim$ 0.2 m·s<sup>-1</sup>) and mainly oriented towards the southeast (Fig. 7i).

**Figure 7.** Contour plots of currents measured at (a-b) the continental shelf (glider transect) and at the Cap de Creus Canyon during (d-e) T1 transect and (g-h) T2 transect (FARDWO-CCC1 cruise). Eastward and northward current velocities are positive, while westward and southward current velocities are negative. Panels (c, f, and i) display stick plots representing the depth-averaged currents associated with dense waters for the three transects, defined between the 28.9 and 29.05 kg·m<sup>-3</sup> isopyenals.

#### 5. Discussion

#### 5.1. Forcing conditions during winter 2021-2022

The winter of 2021-2022 was characterized by two distinct periods: a first typical winter phase from December to January with several short northerly and northwesterly windstorms that caused continuous heat losses in the GoL followed by a second period from February to March dominated by marine storms and lower heat losses (Fig. 3a). During the first period, the averaged sea-atmosphere heat loss was around -158 W·m<sup>-2</sup> (Fig. 3a). In contrast, the second period showed reduced mean

heat losses (-70 W·m<sup>-2</sup>) (Fig. 3a). Overall, averaged sea-atmosphere heat loss for the whole winter was -150 W·m<sup>-2</sup> (Fig. 3a), slightly lower than typical mild winters in the GoL (-200 W·m<sup>-2</sup>) and much lower than severe winters such as 2004-2005 (-300 W·m<sup>-2</sup>) (Schroeder et al., 2010). Despite this, intense short episodes of heat loss (-500 W·m<sup>-2</sup>) occurred in January 2022 (Fig. 3a), which likely contributed to significant surface water cooling and buoyancy loss, especially on the inner shelf (Figs. 3e-g). Water temperature dropped below 12.9 °C in late December/early January (Figs. 3e) and below 12 °C in mid/late February (Figs. 3e), while shelf water density increased to 28.9-28.95 kg·m<sup>-3</sup>, exceeding the pre-winter maximum of 27.9 kg·m<sup>-3</sup> (Fig. 8a). This allowed shelf waters formed over the GoL's shelf to spread as a near-bottom layer across the mid and outer shelf and cascade downslope (Figs. 3 and 8b, c). This likely marked the beginning of the DSWC period, a scenario consistent with previous studies in the GoL (Dufau-Julliand et al., 2004; Durrieu de Madron et al., 2005; Canals et al., 2006; Ulses et al., 2008a).

**Figure 8.** Bathymetric maps of the Gulf of Lion (GoL) showing the potential density anomaly at bottom for four days during the winter of 2021-2022: (a) December 30, 2021; (b) February 25, 2022; (c) March 2, 2022; and (d) April 5, 2022. Density contours (black) are displayed every 0.1 kg·m<sup>-3</sup>. Data has been obtained from the Mediterranean Sea Physics Reanalysis product (Escudier et al., 2020; 2021).

Moreover, during winter, especially throughout February and March 2022, the region was impacted by several easterly/southeasterly windstorms, often associated with increased freshwater discharge from the Rhône River and the coastal rivers (Fig. 3d). Such freshwater inputs are commonly observed during mild winters in the region and can locally reduce salinity and density over the shelf (Ulses et al., 2008a; Martín et al., 2013; Mikolajczak et al., 2020). In contrast, winters favorable to IDSWC, such as 2004-2005, 2005-2006, and 2011-2012, are marked by stronger heat losses, sustained cooling and notably low river discharge (Ulses et al., 2008b). For example, during winter 2004-2005, the Rhône River discharge averaged only ~1400 m³·s¹ between December and March, which is below the mean annual discharge (1920-2007) of 1850 m³·s¹ (Pont et al., 2002; Provansal et al., 2014). These conditions allowed shelf waters to gain potential densities exceeding 29.1 kg·m⁻³, which are sufficient to break through the Eastern Intermediate Water (EIW) layer and generate deep cascading,

even reaching depths > 2000 m (Font et al., 2007; Durrieu de Madron, 2013; Palanques and Puig, 2018). Contrary, during the mild winter of 2021-2022, although winter cooling led to shelf water densification (~28.95 kg·m<sup>-3</sup>), dense waters were not dense enough to overcome the EIW and reach the deep basin. Instead, they spread along the mid-shelf and upper slope (Fig. 8b, c). The cyclonic circulation on the western part of the shelf, driven by the prevailing winds (Ulses et al., 2008a; Bourrin et al., 2008), promoted the transport of these dense shelf waters along the coast (Fig. 7b). The narrowing of the shelf near the Cap de Creus Peninsula further accelerated the overflow of dense shelf waters into the upper slope, mainly cascading into the westernmost canyons (i.e., Lacaze-Duthiers and Cap de Creus Canyons) (Ulses et al., 2008a).

Nevertheless, the mild atmospheric conditions during winter 2021-2022 prevented the transport of dense waters into the deeper sections of these canyons. In fact, the mooring data acquired in the Lacaze-Duthiers and Cap de Creus canyons clearly indicate that dense shelf waters did not reach the deeper canyon sections during that winter. In both moorings, the recorded current speeds remained below  $0.2~\mathrm{m\cdot s^{-1}}$ , temperatures were nearly constant (13.5 °C), and SPM concentrations were < 1 mg·L<sup>-1</sup> (Fig. 4), indicating the absence of a deep cascading. For IDSWC events, previous mooring data have registered much colder near-bottom temperatures (11-12.7 °C), stronger down-canyon velocities (> 0.8 m·s<sup>-1</sup>), and significantly higher SPM concentrations (~40 mg·L<sup>-1</sup>) (Canals et al., 2006).

# 5.2. Dense shelf water cascading in the Cap de Creus Canyon

In early March 2022, during the glider survey, the presence of dense shelf waters (T < 12.9 °C and S < 38.4) was detected at the narrow continental shelf adjacent to the head of the Cap de Creus Canyon (Figs. 5a-c). Previous works have demonstrated that, once dense shelf waters reach the Cap de Creus Peninsula, they are deflected and start to pool the outer shelf until they spill into the Cap de Creus Canyon from the southern flank (Dufau-Julliand et al., 2004; Canals et al., 2006; DeGeest et al., 2008).

The relatively steep bathymetry of the inner shelf, combined with the prevailing southeastward currents (Figs. 7a-c), facilitated the transport of dense waters into the upper section of the Cap de Creus Canyon. Most likely, these dense waters were pushed into the canyon by cascading currents or storm-induced downwelling processes associated with northerly/north-westerly winds (Fig. 3b), as previously documented for the mild winter of 2010-2011 (Martín et al., 2013). Within the upper canyon section, the T1 transect showed a vein of cold, fresher water, rich in oxygen and chlorophyll-a between 100 and 400 m depth (core depth: 380 m) (Figs. 5d-f and 6d-f), above the EIW (~400 m depth) (Fig. A1). This flow traced the export of a vein of coastal dense waters exported from the shelf towards the upper continental slope, which contributed to the body of WIW (Figs. 5d-f and A1). The WIW likely formed on the outer shelf of the GoL during winter through intense surface cooling and convective mixing of the AW, and then advected southward to the Cap de Creus Canyon by the general circulation (Lapouyade and Durrieu de Madron, 2001; Dufau-Julliand et al., 2004; Juza et al., 2019).

If we examine the hydrodynamics of the dense water plume, it exhibits characteristics consistent with the "head-up" configuration described in Shapiro and Hill (2003), in which most of the dense water remains upslope while only a thin tail extends downslope. This configuration suggests a limited downslope export of dense waters. To further characterize the flow regime of the plume, we estimated two dimensionless numbers: the Richardson number (Ri) and the Froude number (Fr), both of which provide insights into the stability and dynamical behaviour of stratified flows.

The Ri is a ratio between the Brünt-Väisäla frequency squared (N<sup>2</sup>) and the vertical shear of the horizontal velocity (S<sup>2</sup>), and describes the tendency of a stratified flow to remain laminar or become turbulent (Monin and Yaglom, 1971). For this calculation, we used temperature, salinity, and E-W current velocities from CTD and LADCP measurements acquired during the cruise. Ri values showed a general increase between 150 and 300 m depth, which roughly corresponds to the vertical extent of the dense shelf water plume (Fig. C1). The maximum Ri observed reached 0.18 at 270 m depth in the upper canyon, and 0.16 at 180 m in the mid canyon. These values are below the critical threshold of 1 that separates laminar (Ri >1) from

turbulent flow regimes, thus indicating a predominantly turbulent flow (Mack and Schoeberlein, 2004). This suggests that fluid instabilities likely enhanced vertical mixing and lateral spreading of the dense water plume.

Complementary, we estimated the Froude number  $(Fr = U/\sqrt{g'h})$ , following the approach by Cenedese et al. (2004), which distinguishes between laminar, wavy, and eddying flow regimes in dense bottom currents over slopes. Using a representative current velocity (U) of  $0.18 \text{ m} \cdot \text{s}^{-1}$ , a dense layer thickness (h) of 100 m, and a reduced gravity (g') of  $2.67 \cdot 10^{-4} \text{ m} \cdot \text{s}^{-2}$ , we obtained Fr ~1.10. This value lies just above the critical threshold of 1, indicating a supercritical flow regime where inertial forces become more significant, potentially favoring more unsteady and turbulent flow conditions (Cenedese et al., 2004).

Within the Cap de Creus Canyon, coastal dense shelf waters flowed predominantly along the southern canyon flank, associated with increased SPM concentrations (Figs. 6 and 7). This suggests that during this MDSWC event, dense waters transported sediment particles into the canyon down to intermediate depths (Fig. 6d). The time series of water discharge from the Rhône River and the smaller coastal rivers does not seem to indicate important freshwaters and sediment inputs prior to the cruise (Fig. 3d). In fact, both the Rhône River and the coastal rivers had relatively low discharges before the cruise (Fig. 3d). This points to storm-waves and cascading-induced currents, rather than riverine input, as the main mechanisms of sediment resuspension and transport. This is consistent with previous findings by Estournel et al. (2023), who described that, during the mild winter of 2010-2011, storm-induced downwelling events contributed to the remobilization of shelf sediments that fed the upper section of the Cap de Creus Canyon and along the North Catalan shelf. In fact, during mild winters, most of the sediment delivered by the GoL's rivers remains relatively close to their mouths after floods and it is later remobilized by storm waves under highly energetic conditions (Guillén et al., 2006; Estournel et al., 2023). In addition, the increased fluorescence values observed within the dense water plume suggests the mixing of terrestrial sediments with phytoplankton cells (Figs. 6e, h). Phytoplankton blooms, which typically peak between December and January, likely contributed newly produced biological particles to GoL's shelf waters, thereby enhancing the transport of organic matter into the canyon during the observed MDSWC event (Fig. B1; Fabres et al., 2008).

The dense water plume continued flowing down-canyon to the mid-canyon section, where it contributed to the WIW body that extended to depths of 380 m (Figs. 5g-i). At these depths, enhanced SPM concentrations were observed on the southern flank, contrasting with clear waters on the northern canyon flank (Fig. 6g). The contrast in SPM concentrations between the flanks of the Cap de Creus Canyon actually reflects its marked asymmetric morphology and hydrodynamics. One key aspect is the very narrow adjacent shelf (2.5 km), which brings the southern rim of the canyon into close proximity with the coast (Lastras et al., 2007; Durán et al., 2014). The canyon itself presents very dissymmetric walls: the northern canyon wall is relatively smooth, with fine-grained silty clays and relatively high accumulation rates (up to 4.1 mm·yr<sup>-1</sup>), indicative of a depositional area. On the other hand, the southern flank has erosional features (exposed bedrocks and gullies), and coarser sediments such as gravel. These characteristics, along with lower accumulation rates (~0.5 mm·yr<sup>-1</sup>), suggest active sediment transport and bypassing processes (Lastras et al., 2007; DeGeest et al., 2008). In addition, other studies have evidenced the presence of furrows oblique to the canyon axis, which have been interpreted as signatures of DSWC events over the southern canyon wall (Canals et al., 2006). This has been later confirmed through modeling and observations (DeGeest et al., 2008), which also identified episodic strong currents in the southern canyon rim during cascading events in this area. Furthermore, the conceptual model proposed by DeGeest et al. (2008) suggest that the coastal current forms an eddy over southwestern shelf near the Cap de Creus headland (Davies et al., 1995), which likely steers dense water flows and sediment preferentially toward the southern canyon rim (DeGeest et al., 2008). Altogether, these morphological and hydrodynamic features offer a plausible explanation for the enhanced SPM concentrations observed along the southern canyon wall in both upper- and mid-canyon sections during our study (Figs. 6d, g).

Our data also show that dense waters along the southern canyon flank shifted from a predominantly southeastward direction in the upper-canyon section to a more southward direction in the mid-canyon section (Figs. 7f, i). This transition

suggests that, as the canyon topography opens, dense waters are no longer confined by its walls and begin to veer southward, following the continental slope (Millot, 1990). Previous studies have shown that part of these waters can continue flowing southward crossing the Gulf of Roses and reaching the neighbouring Palamós and Blanes Canyons (Zúñiga et al., 2009; Ribó et al., 2011). This is supported by the presence of a shallow NNE-SSW oriented channel that extends from the narrow shelf of the Cap de Creus Canyon to the head of the Palamós Canyon (Durán et al., 2014). During MDSWC events, this channel facilitates the along-slope transport of dense waters (particularly WIW) and associated suspended sediment towards the Palamós Canyon (Ribó et al., 2011; Martín et al., 2013). A similar mechanism may have occurred during the studied winter here (2021-2022), when WIW could have been advected southward through this channel and eventually reached neighbouring canyons, as previously documented during the winter of 2017 (Arjona-Camas et al., 2021).

#### 5.3. Spatial and temporal variabiliaty of dense shelf water export in the Cap de Creus Canyon

At the continental shelf transect (Fig. 1), dense water transport was estimated at 0.7 Sv and 10<sup>5</sup> metric tons of SPM. In the upper canyon section (T1 transect), the transport associated with dense waters was 0.3 Sv, including 10<sup>5</sup> tons of SPM. These values are comparable to those observed during the mild winter of 2010-2011 in the Cap de Creus Canyon, for which Martín et al. (2013) estimated a total transport of 10<sup>5</sup> metric tons of SPM though the southern canyon wall. Further downslope, at the mid-canyon section (T2 transect), transport across the southern flank decreased to 0.05 Sv, with roughly 10<sup>4</sup> metric tons of SPM. This transport reduction between the upper and mid-canyon sections suggests that most of the dense water either remained on the shelf, skirted around the Cap de Creus headland, or cascaded only into the upper canyon. Part of the dense shelf waters could have also followed the shallow channel located between 100 and 150 m depth, as previously discussed (see Fig. 4 in Durán et al., 2014).

Since our observations provided only a snapshot of this MDSWC event, we used reanalysis data to determine its temporal extent and to place it in the broader context of cascading events in the Cap de Creus Canyon over the past two decades. We assessed the reliability of the MedSea by comparing it with our CTD observations. In particular, we compared depth-averaged temperatures within the dense water plume (150-390 m depth) at stations where dense waters (T< 12.9 °C) were detected, using the root mean square error (RMSE) statistical method. The RMSE errors were consistently low across all stations (< 0.2 °C) (Table 1), indicating a good agreement between both datasets. These results, along with the detailed validation presented in Fos et al. (2025), support the use of the MedSea reanalysis product for studying DSWC in this region. In that study, the authors compared the MedSea output with long-term mooring data from the Cap de Creus and Lacaze-Duthiers canyons, and showed that, at 1000 m depth, the MedSea reproduces 84% of individual DSWC events within the same week and 56% on the exact date (Fos et al., 2025).

**Table 1.** Comparison of depth-averaged temperatures within the dense water plume derived from observations and reanalysis, with corresponding root mean squared errors (RMSE).

| Stations | Observed<br>temperature (°C) | Reanalysis<br>temperature (°C) | RMSE (°C) |
|----------|------------------------------|--------------------------------|-----------|
| T1-01    | 12.41                        | 12.42                          | 0.17      |
| T1-02    | 12.55                        | 12.46                          | 0.18      |
| T1-03    | 12.34                        | 12.42                          | 0.19      |
| T1-04    | 12.76                        | 12.42                          | 0.21      |
| T2-01    | 12.96                        | 12.71                          | 0.15      |
| T2-02    | 12.68                        | 12.65                          | 0.04      |
| T2-03    | 12.72                        | 12.67                          | 0.05      |

Daily reanalysis data indicated that the export of dense water was consistently higher in the upper-canyon section than in the mid-canyon section throughout winter 2021-2022 (Fig. 9a), consistent with observational results. A significant

increase of dense water export occurred in late January (0.2 Sv), coinciding with a drop in temperature (T < 12.9 °C) and a slight decrease in potential density to 28.85 kg·m<sup>-3</sup> (Fig. 9). Although an increase in transport was also detected in the mid-canyon section (Fig. 9a), it was notably smaller in magnitude (~0.12 Sv), and the potential temperature time series do not seem to indicate the presence of dense shelf waters in the canyon at that depth (Fig. 9b). Three shallow cascading pulses were further identified in mid- and late February and early March, which had comparable down-canyon transport magnitudes (~0.2 Sv) in the upper canyon section (Fig. 9a). The mid-February pulse was particularly evident in the upper canyon section, which coincided with a temperature drop below 12.5 °C (Fig. 9b) that indicated the arrival of dense waters at this depth. Another shallow cascading was identified in mid-March, representing the strongest transport event in the canyon.

This mid-March event was particularly noted in the upper-canyon section (with ~0.4 Sv) and, to a lesser extent, in the mid-canyon section (~0.1 Sv) (Fig. 9a). The time series of temperature (~12 °C) and the concurrent increases in potential density (~28.87 kg·m<sup>-3</sup>) (Figs. 9b, c) also confirmed the arrival of dense waters at both locations. The enhanced export observed in mid-March was likely triggered by the easterly/southeasterly storm on March 13 (Figs. 3b, c), which promoted MDSWC into the Cap de Creus Canyon. The role of marine storms as drivers of downwelling and cascading events has been previously documented for other mild winters, such as 2003-2004 and 2010-2011 (Ulses et al., 2008a; Martín et al., 2013). Nonetheless, these transport peaks are lower than the peak transport estimated in the Cap de Creus Canyons during the extreme winter of 2004-2005 for intermediate waters (~0.6 Sv) (Fig. 10), and for deep waters (1.29 Sv) (Fos et al., 2025).

As March progressed, and net heat losses decreased (Fig. 3a), dense water export gradually decreased, eventually ceasing in early April (Fig. 9a). Accordingly, the duration of the cascading period in the Cap de Creus Canyon, defined as the period between the first and last cascading event (Palanques et al., 2006), was approximately two months, from late January to mid-March 2022 (Fig. 9). During this winter, shallow cascading events primarily affected the upper canyon, with a weaker influence further downcanyon in the mid-canyon section (Fig. 9a). Indeed, the total volume of dense water exported during the cascading period in winter 2021-2022 was estimated at 260 km<sup>3</sup> in the upper canyon section, compared to approximately 11 km<sup>3</sup> in the mid-canyon section.

**Figure 9.** (a) Time series of along-canyon transport (Sv) of dense waters, (b) potential temperature (°C), and (c) potential density (kg·m<sup>-3</sup>) for the upper-canyon transect (dark blue) and the mid-canyon section (light blue), extracted from the Mediterranean Sea Physics Reanalysis product (Escudier et al., 2020; 2021). In panel (a), positive values indicate down-canyon export, while negative values indicate up-canyon export.

# 5.4. Interannual variability of dense shelf water transport and future perspectives of DSWC

Over the past two decades, MDSWC events have been the most common form of dense shelf water export through the Cap de Creus Canyon, as reported in previous studies (Herrmann et al., 2008; Durrieu de Madron et al., 2023), and as further evidenced by the time series of dense water transport estimated from reanalysis data shown in Figure 10. In contrast, IDSWC events have occurred less frequently, typically every 5-7 years, and are often associated with higher transports of dense shelf waters (Fig. 10).

Putting the winter 2021-2022 into perspective, the total dense water transport (260 km<sup>3</sup>) was approximately half of the total volume simulated for the mild winter of 1998-1999 (500 km<sup>3</sup>) (Dufau-Julliand et al., 2004), and higher than the total export estimated for the mild winter of 2003-2004 (75 km<sup>3</sup>) (Ulses et al., 2008a). Compared to the extreme IDSWC event of 2004-2005, one of the most studied events in the GoL, the 2021-2022 MDSWC event accounted for roughly half of the total dense water volume exported through the Cap de Creus Canyon (> 750 km<sup>3</sup>) (Canals et al., 2006; Ulses et al., 2008b). However, as shown in the reanalysis time series (Fig. 10) and recent numerical simulations (Mikolajczak et al., 2020), mild and extreme winters can sometimes reach similar magnitudes of dense water export despite having contrasting atmospheric conditions. For example, during the mild winter of 2010-2011, the total dense water export (1500 km<sup>3</sup>) was comparable to the extreme winter of 2004-2005 (1640 km<sup>3</sup>) (Ulses et al., 2008a). The key difference lies in the preferential export pathways: in 2004-2005, 69% of dense shelf water cascaded through the Cap de Creus Canyon down to the deeper parts of the canyon. In contrast, in winter 2010-2011, only 30% of the export occurred through the canyon, while 70% followed along the coast or remained around the upper canyon section (Ulses et al., 2008a; Mikolajczak et al., 2020). This difference may also affect the distribution of suspended sediment, which instead of being advected to deeper canyon areas, is likely redistributed along the shelf or accumulates in the upper canyon. Based on our results, it is very likely that during the winter of 2021-2022, a higher proportion of dense waters and associated suspended sediment was exported along the coast or only generated shallow cascading into the upper canyon, resembling the export pattern observed during the mild winter of 2010-2011. These findings suggest that during mild winters, MDSWC events may primarily influence coastal ecosystems, whereas in extreme winters, IDSWC may significantly impact deep-benthic ecosystems (Mikolajczak et al., 2020).

**Figure 10.** Time series of daily down-canyon transport of dense shelf waters (Sv) from 1997 to 2022, calculated across a central transect in the upper region of the Cap de Creus Canyon (see location in Figure 1). The occurrence and magnitude of IDSWC events is highlighted in red, whereas mild events are highlighted in blue. Data have been extracted from the Mediterranean Sea Physics Reanalysis product (Escudier et al., 2020; 2021).

Future climate projections indicate an increasing frequency of mild winters in the northwestern Mediterranean (Herrmann et al., 2008; Durrieu de Madron et al., 2023). Under the IPCC-A2 scenario, DSWC could decline by 90 % at the end of the 21<sup>st</sup> century (Herrmann et al., 2008). This scenario would drastically reduce both the intensity and depth penetration of DSWC.

Recent studies have already pointed out to a warming and salinification of surface and intermediate waters across the northwestern Mediterranean (Margirier et al., 2020), which would increase the stratification of the water column and hinder deep-ocean convection. Such reduction could strengthen the production of WIW by favouring intermediate-water formation over deep-water ventilation (Parras-Berrocal et al., 2022). Consequently, dense shelf water transport could become more limited to near-coastal areas or upper-canyon regions, altering the regional hydrology and impacting the Mediterranean Thermohaline Circulation (Somot et al., 2006).

Ongoing sea warming and increasing stratification are also expected to impact shelf-slope exchanges, reducing IDSWC activity and the associated transport of particulate matter from the GoL to the deep basin (Somot et al., 2006). As a result, the transfer of dense shelf waters and organic matter to the deep sea would be drastically reduced and mainly redirected along the Catalan shelf or the upper reaches of the Cap de Creus Canyon (Estournel et al., 2023). A decline in the frequency of marine storms could also promote sediment retention in the Rhône River prodelta, reducing the transfer of particulate matter towards the slope and deep basin (Estournel et al., 2023). These changes could significantly alter transport pathways and affect erosion and deposition patterns along the incised submarine canyons of the GoL, and in consequence, have important impacts in deep benthic communities that rely on the arrival of suspended particulate matter. Cold water corals, benthic invertebrates, and commercially important species such as shrimps may be particularly vulnerable, as the limited supply of nutrients and oxygen would hinder their survival (Puig et al., 2001; Pusceddu et al., 2013). In Lacaze-Duthiers Canyon, for instance, Chapron et al. (2020) showed that cold-water coral growth is strongly influenced by the intensity of DSWC events, which modulate their feeding conditions and development. Intense DSWC events cause high budding rates but low colony linear extension by limiting prey capture rates with high current speeds. Mild DSWC events cause high budding rates and high linear extension associated with higher organic matter supply. In contrast, the absence of cascading plumes can cause high mortality (Chapron et al., 2020). Regardless of the direction of DSWC's effects on deep-sea ecosystems, it has been demonstrated that even a small loss of biodiversity can lead to a major ecosystem collapse (Danovaro et al., 2008), with cascading impacts on ecosystem services such as fisheries (Pörtner and Knust, 2007; Smith et al., 2009).

#### 6. Conclusions

This paper provides a comprehensive observational characterization of shelf-slope exchanges of dense shelf waters and associated SPM in the Cap de Creus Canyon during a MDSWC event. We combined glider data, ship-based CTD transects, instrumented mooring lines, and reanalysis data to investigate dense water and sediment export during the mild winter of 2021-2022.

The first part of winter 2021-2022 (December and January) was characterized by several days of northerly and north-westerly winds, which induced sustained heat loss, surface cooling, and densification of surface waters over the GoL. In contrast, easterly and southeasterly winds during the second part of the winter (February and March) enhanced the export of dense shelf waters to the southwestern end of the gulf, where they cascaded into the upper section of the Cap de Creus Canyon. In early March 2022, dense shelf waters were observed at the continental shelf during the glider survey. These waters cascaded into the upper canyon down to ~390 m depth, where they contributed to the WIW body. Increased SPM concentrations were also observed at the same water depths, likely indicative of a resuspension process. River discharges were low before and during the cruise and, therefore, they were not the main source of suspended sediments. Instead, storm-induced currents

associated with cascading likely resuspended and transported suspended sediments through the canyon. We estimated an export of dense water of  $\sim$ 0.3 Sv and 10<sup>5</sup> metric tons of sediments in the upper canyon section.

Reanalysis data provided additional temporal context, revealing that the cascading period lasted approximately two months, from late January to mid-March 2022. These data showed several shallow cascading events during this period, with the peak export in mid-March associated with an easterly/southeasterly storm that likely pushed dense shelf waters to the canyon. Over the entire cascading period, we estimated a total dense shelf water export of ~260 km³. In terms of volume, the 2021-2022 export was approximately half of that estimated for the mild winter of 1998-1999 and higher than the export simulated for the mild winter of 2003-2004. Compared to the extreme winter of 2004-2005, it represented roughly half of the total dense water volume exported. These contrasts show the episodic and variable nature of DSWC in the GoL. Moreover, it is likely that, during the winter of 2021-2022, a high proportion of dense waters and associated SPM was exported along the coast or generated only shallow cascading into the canyon, resembling the export patterns of previously reported MDSWC events.

Future research should benefit from a more detailed exploration of the physical dynamics driving DSWC. More importantly, given the climate change scenario, future studies should focus on the evolution and changes on the WIW. These studies would help to understand the alterations in the water column properties and changes in stratification affecting convection processes, which directly impact the Mediterranean thermohaline circulation and the climate system.

# Data availability

733

- The data that support the findings of this study are publicly available under the following links: SeaExplorer glider from
- (https://data-selection.odatis-ocean.fr/coriolis/uri/p83112098), CCC-1000 moored time series
- (https://doi.org/10.17882/104746; Sanchez-Vidal et al., 2025a), LDC-500 and LDC-1000 moored time series
- (https://doi.org/10.17882/45980; Durrieu de Madron et al., 2024). The CTD and the ADCP data collected during the
- FARDWO-CCC1 Cruise are available at <a href="https://doi.org/10.17882/105499">https://doi.org/10.17882/105499</a> (Sanchez-Vidal et al., 2025b).

# 727 Authors contributions

- AS, DA, XD, and FB defined the research problem, the conceptualization of the study, and leaded the acquisition of the study
- data. MA-C carried out the data analysis and produced the figures and first draft of the manuscript. All coauthors discussed
- the analyses and contributed to the review and writing of the final paper.

# 731 Competing interests

The authors declare that they have no conflict of interest.

# Acknowledgements

- We are very grateful to the captain, crew, and the scientific team for their dedication and hard work during the FARDWO-
- CCC1 Cruise at the R/V García del Cid, and to the technicians for their guidance and assistance. We wish to thank the Alseamar
- Alcen technicians for the glider deployment and piloting, and the data pre-processing. We also want to thank Pascal Conan
- from OSU-Stamar (Stations Marines UPMC de Banyuls-sur-Mer) at the Sorbonne Université, as well as acknowledge the
- MOOSE, the COAST-HF and the SOMLIT programs coordinated by CNRS-INSU and the Research Infrastructure ILICO
- (CNRS-IFREMER) for providing data from the POEM and SOLA buoys and the Lacaze-Duthiers Canyon's mooring.

# 740 Financial support

- This work has been supported by the FARDWO (PID2020-114322RB-I00) project funded by
- MICIU/AEI/10.13039/501100011033, the Catalan Government Excellent Research Groups gran no. 2021 SGR 01195, and the

- ANR MELANGE (ANR-19-ASMA-0004) project. MAC was supported by a Margarita Salas postdoctoral grant (2022-2024)
- from the Spanish Ministry of Universities throughout part of the project. HF is supported by the AGAUR FI-SDUR fellowship
- number 2023-FISDU-00233 by Secretaria d'Universitats i Recerca from the Catalan Government.

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

**Figure A1.** TS diagram from the CTD profiles collected during FARDWO-CCC1 T1 and T2 transects across the Cap de Creus Canyon, as well as those collected during the glider survey at the adjacent shelf (see Fig. 1b for positions). The different water masses that can be identified in the study area are: dense shelf waters, oAW (old Atlantic Water), WIW (Western Intermediate Water), and EIW (Eastern Intermediate Water).

Appendix B – Seasonality of chlorophyll-a between October 2021 and April 2022 from in situ measurements at the SOLA station.

**Figure B1.** Seasonality of the chlorophyll-a measured at SOLA station between October 2021 and April 2022. Green dots indicate the monthly maximum values. Data has been retrieved from the SOMLIT-SOLA monitoring site in the Bay of Banyuls-sur-Mer (<a href="https://www.seanoe.org/data/00886/99794/">https://www.seanoe.org/data/00886/99794/</a>; Conan et al., 2024).

Appendix C – Richardson number (Ri) profiles calculated for selected stations during the FARDWO-CCC1 cruise in the Cap de Creus Canyon in March 2022

**Figure C1.** Vertical profiles of the Richardson number (Ri) calculated from CTD and LADCP measurements at two locations at the Cap de Creus Canyon where dense water were observed: T1-05 (upper canyon transect, dark blue) and T2-03 (mid-canyon transect, light blue). Values Ri <1 indicate a turbulent regime.