# Peer review of "Dense shelf-water and associated sediment transport in the Cap de Creus Canyon and adjacent shelf under mild winter regimes: insights from the 2021-2022 winter"

_EGUsphere, 2025_

## Referee Comment (RC2)

This study addresses the dense-shelf water and associated sediment transport in the Cap de Creus Canyon during the mild winter of 2021-2022. This canyon has been identified as a main pathway for the transfer of dense shelf water and sediments from the shelf to the slope and deep margin. The study bases on combination of data from gliders, ship-based CTD transects, instrumented mooring lines, and a reanalysis product.

The article is very clearly written and organized. The results are supported by a set of observations covering different spatio-temporal scales, which is an asset. I do no have any problem with the manuscript other than it is a bit hard to follow because of its very descriptive nature given the different datasets involved. In contrast, I think that the relevance of the study is not very clearly stated. However I do not know the region very well, so I ignore the state of the scientific knowledge and the reach of the relevance or novelty of this study, so I prefer not to evaluate that point.

Overall, it is a good paper. Mi main criticism is about the possibilities that the use of the reanalysis product offers, and which I feel it's not exploited. I wonder why not to (really, with numbers) validate this reanalysis with your observations, and make the same computations for several years, separating mild and intense winter conditions. This would greatly strengthen the paper's conclusions. So far, the article is a very nice compilation of observations from different datasets, but it is very descriptive and the cause-effect of the findings is often weakly sustained. I really think there is potential for more robust conclusions if further analysis were carried out by adding a longer time series from the reanalysis to put this winter, and other mild winters in context. This would allow to generalize your conclusions.

**General comments**

Abstract :
I didn't really understand if the Cap de Creus Canyos is "**only a partial sink** of cascading waters" or if "remarkable dense shelf water and sediment transport occurs in the Cap de Creus Canyon,..., **even during mild winters** ". Isn't this a bit contradictory? Or maybe I'm missing the difference between these transports. In any case, please clarify. This is a question that remained even after reading the full manuscript.

Methods:
The interpolation method used in the sections should be stated. The figures look a bit weird and I think it might be an interpolation issue.

**Particular comments**
L51. What "it" makes reference to?

L74-75. More prevalent than the extreme ones, thus, reducing overall DSWC over time? Or more prevalent than the "no DSWC scenario", thus, increasing overall DSWC over time?

L99-101. I'd remove " *which was monitored during the FARDWO-CCC1 cruise, and simultaneous measurements at its adjacent shelf acquired during a glider survey as part of the MELANGE-DUNES experiment.*" from here as it's too much detail for the introduction.

L118. Export of what? Just precise

L129. What do you mean with "the concentration of water"? Are you refering to the residence time? Please rewrite, the term is awkward.

L.136. The full water column gets mixed? It would be surprising.

L.149. 300-400 m is the upper limit I guess, above which stratification prevents the full mixing of the water column? In that case that would rather be a re-stratification, because DSW forms from the surface forcing, and the a light water layer develops in the surface. Is that it?

L.151. Gain

L.164-165. However, all the point of TEOS10 is to promote the use of the more adequate conservative temperature and absolute salinity instead.

L.193-194. But what's the range of the bottom depth?

L.216. Data is a plural noun: "Data were.."

L.226-228. What type of data were used? Is it discharge volume?

L.286. Low compared to what? Give a reference please.

L.287. That's kind of surprising the existence of a storm that is not cause by strong winds, isn't it? Can you provide an explanation?

L292. This is also surprising!

L.293. Low compared to which reference value?

Fig 3. It would be better to inverse the y-axis for density, so the densest water corresponds to the bottom layers.

L319 and throughout the manuscript. It would be better to refer to the Moose stations by their location instead of LDC or CCC, which is complicated to remember.

L.336. Compared to what reference values? (please provide references whenever you state that XX values are low or high).

Fig 5. Please avoid the used of divergent color maps for non-divergent fields as in the left column. This is misleading. Also, I'd personally prefer to see latitude instead of distance in the x-axis. I think it helps the readers to know where they are.
L341. This information belongs to methods. I actually missed it when I read it.

L.340-350. I suggest to better indicate what is from glider and what from cruise. It took me a moment to understand.

Fig 6: The color bars for panels f and i are not the same, even if they have the same limits and correspond to the same variables, which is misleading and makes comparison difficult.

L430. However, the discharge was low this winter, and dense water forms other years. This makes me think that this is not a reason to justify the low density.

L.432-435. I can't really see a decrease in density, which makes me think that river discharge is not a key factor.

Fig 8. Wouldn't it be better to plot bottom density in order to identify dense water? Also, please change the color map for a non-divergent one. This one is misleading.

L.446-447. As I said above, we cannot judge if the values are low or high if we don't have references.

L479. Suggest.

L489. Flows.

L.500-510. This paragraph should definitely go to Methods and not in the discussion.

L.513. 0.05 Sv is practically zero, taking into account the strong variability. I actually would say the mean is negative? Have the authors double checked this mean? In any case, given the difference in the T1 and T2 value, I would not define the Cap de Creus Canyon as a partial sink, it is rather not at sink during mild winters. Wether or not this canyon is a sink, or export occurs through it remains confusing to me throughout the manuscript.

L519-520. You state you used the reanalysis "to assess the variability of dense shelf water export in the Cap de Creus Canyon during the mild winter of 2021-2022." but the computation spans the October-May period, so, beyond winter.

L.525. I miss having some numbers to compare the reanalysis with the observations and quantify how well they match. You should plot the same variable for the T1 and T2 transects, integrated over the same depths. You could event add a line for the value of each variable in your observations. This would provide robustness to the reanalysis results.

L.546. "relatively weak wind forcing".

L.560-562. How was this percentage estimated? I'm a bit confused. When we say export, I think about the water transport down-canyon to reach deeper depths, if water doesn't get to leave the shelf I wouldn't call it export. Throughout the manuscript the authors state (and the transport numbers suggest) that the actual export is very weak. I would like to know how these percentage were computed and, as asked before, what are the reference values in Sv (for instance a climatological mean, or the typical values in strong winters) for transport.

---

## Author Comment (AC1)

**Response to reviewer 2**

Dear reviewer,

We thank you very much for your constructive and relevant comments to our manuscript. Below, your reviews are reproduced in **black** font and our responses interspersed in **blue**.

Since the reviewer #1 also raised important points, we kindly suggest to take a look at our responses to Reviewer #1 as well.

Please, note that all **line numbers** in our responses refer to the clean version of the manuscript, not the tracked-changes version.

This study addresses the dense-shelf water and associated sediment transport in the Cap de Creus Canyon during the mild winter of 2021-2022. This canyon has been identified as a main pathway for the transfer of dense shelf water and sediments from the shelf to the slope and deep margin. The study bases on combination of data from gliders, ship-based CTD transects, instrumented mooring lines, and a reanalysis product.

The article is very clearly written and organized. The results are supported by a set of observations covering different spatio-temporal scales, which is an asset. I do not have any problem with the manuscript other than it is a bit hard to follow because of its very descriptive nature given the different datasets involved. In contrast, I think that the relevance of the study is not very clearly stated. However, I do not know the region very well, so I ignore the state of the scientific knowledge and the reach of the relevance or novelty of this study, so I prefer not to evaluate that point.

**Reply:** We appreciate the recommendation for minor revisions. However, we have thoroughly revised the manuscript, addressing all your comments in detail as if it had been a major revision. Your revisions have been very helpful in improving the clarity and strength of the manuscript, as well as in preparing a more focussed discussion and better contextualize our findings.

We have reorganized both the Introduction and Discussion sections to make the paper message more concise and impactful (see modified sections in the revised version of the manuscript). In this regard, we would like to briefly emphasize the novelty and significance of our study.

Most previous studies in the Cap de Creus Canyon (and more broadly in the Gulf of Lion) have primarily focused on intense dense shelf water cascading events (IDSWC). These events are more energetic and have greater effects, making their impacts easier to quantify. This explains the significant attention they have received over the past decades (e.g., Heussner et al., 2006; Canals et al., 2006; Puig et al., 2008; Ogston et al., 2008; Sanchez-Vidal et al., 2008; Durrieu de Madron et al., 2013). However, IDSWC are not the most frequent in the region. In contrast, mild dense shelf water cascading (MDSWC) events have been more common since the beginning of the observational era in the Gulf of Lion and are expected to become more prevalent under climate change scenarios (Herrmann et al., 2008).

Previous work, such as Ulses et al. (2008a), Martín et al. (2013), Rumín-Caparrós et al. (2013), or Mikolajczak et al. (2020), have provided valuable insights into the dynamics of MDSWC in the Cap de Creus Canyon. These studies have mostly relied on mooring time series acquired in the canyon head and/or model outputs to detect the presence of dense water and infer their export pathways. However, they do not offer a comprehensive observational characterization of the hydrographic properties of the water column, current dynamics, or the shelf-to-canyon export of dense waters during these events.

Our study builds on these studies and complements these papers by integrating a comprehensive observational dataset, which includes concurrent observations at the Cap de Creus Canyon and the adjacent continental shelf, with reanalysis data to analyze in detail a recent mild winter (2021-2022). We

document in situ the presence of cascading waters in the canyon, which contribute to the body of Western Intermediate Water (WIW), at different locations from the shelf to the mid-canyon. In addition, and as recommended by the reviewer, we place this winter in a multi-winter context thank to reanalysis data, which allows us to compare it with other mild and intense winters over the past two decades.

To our knowledge, and thanks to the efforts of the FARDWO project and the MELANGE-DUNES experiment, this is the first time that a MDSWC event in the Cap de Creus Canyon has been characterized in such detail based on direct observations spanning the upper and mid sections of the canyon, as well as its adjacent shelf. We believe that, by addressing a relatively understudied but increasingly relevant phenomenon, our work fills an important knowledge gap and contributes to a better understanding of how moderate winters affect DSWC events in the Cap de Creus Canyon, and how these conditions may affect shelf-slope exchanges, WIW formation, and sediment transport pathways in the future. These are all crucial aspects for anticipating future changes in canyon functioning and deep-sea ecosystems.

Overall, it is a good paper. My main criticism is about the possibilities that the use of the reanalysis product offers, and which I feel it's not exploited. I wonder why not to (really, with numbers) validate this reanalysis with your observations, and make the same computations for several years, separating mild and intense winter conditions. This would greatly strengthen the paper's conclusions. So far, the article is a very nice compilation of observations from different datasets, but it is very descriptive and the cause-effect of the findings is often weakly sustained. I really think there is potential for more robust conclusions if further analysis were carried out by adding a longer time series from the reanalysis to put this winter, and other mild winters in context. This would allow to generalize your conclusions.

Due to this, I think that the paper can be accepted after minor revisions, but it would be a better paper with major revisions.

**Reply:** Thank you for your insightful comment. We fully agree that validating the reanalysis product and placing our observations in a broader temporal context would strengthen the conclusions of our paper.

Regarding the validation, we have not included it in this manuscript because it is the focus of a separate study of our group that we have recently submitted to the same journal (Fos et al., 2025). In that paper, we validate the Mediterranean Sea Physics reanalysis product against long-term mooring observations in both the Cap de Creus and Lacaze-Duthiers canyons. The preliminary results of that paper show that the reanalysis accurately reproduces DSWC events, matching 84% of IDSWC events within the same week and 56% on the exact date. These findings actually reinforce the reliability and applicability of reanalysis data in our study region. Nevertheless, in the Discussion section (5.3.), we have provided the root mean square errors (RMSE) resulting from the comparison between observational and reanalysis data for T1 and T2 transects at depth where dense shelf waters are detected, in order to validate the use of reanalysis data for our paper. We have also made the pertinent comments on the Discussion section.

Furthermore, we acknowledge the value of placing our MDSWC in a longer-term context. In response to your suggestion, we have currently extended the analyses to include a multi-year time series (from 1997 to 2022) of dense water transport through the canyon, based on the same reanalysis product (Fig. 10). This allows us to compare the 2021-2022 winter with previous IDSWC and MDSWC events and support more general conclusions of the variability of this process and how it has changed throughout the years.

**General comments**

**Abstract:**
I didn't really understand if the Cap de Creus Canyon is "**only a partial sink** of cascading waters" or if "remarkable dense shelf water and sediment transport occurs in the Cap de Creus Canyon, **even during mild winters**". Isn't this a bit contradictory? Or maybe I'm missing the difference between these

transports. In any case, please clarify. This is a question that remained even after reading the full manuscript.

**Reply:** We understand that the use of these terms is a bit contradictory. Our intention was to emphasize that, even during mild winters, the canyon still acts as a preferential pathway for the transport of dense shelf waters and associated suspended sediments, although their transport is mostly confined to the upper canyon and, to a lesser extent, to the mid canyon. We have removed these terms throughout the manuscript to avoid confusion.

**Methods:**
The interpolation method used in the sections should be stated. The figures look a bit weird and I think it might be an interpolation issue.

**Reply:** The interpolation method that we have used in the sections is "isopycnic gridding". This method is a gridding procedure that organizes the hydrographic data along surfaces of constant potential density (isopycnals) rather than constant depths. By doing so, it better preserves the vertical structure of water masses and reduces artificial smoothing across density gradients (Schlitzer, 2023).

In our work, the hydrographic profiles obtained from the CTD stations collected during the FARDWO-CCC1 cruise within the Cap de Creus Canyon, as well as those from the glider section, were interpolated onto a regular grid using this isopycnic gridding method.

We have clarified the interpolation method at the corresponding sections (lines 170-173 and 211-212).

Additionally, we have reorganized the Methods section to improve the clarity of the manuscript and ensure that each type of dataset is clearly introduced. In particular, we have incorporated the SOLA station observations into section 3.2.1. alongside with heat fluxes, wind and wave data, and river discharge. We have also renamed this section as "Environmental forcings and shelf observational data". We have also created the section titled "3.4. Estimation of dense water and SPM transports from observations". Here, we describe the methodology used to estimate both the dense shelf water transport (in Sv) and the associated suspended particulate matter (SPM) transport (in metric tons) for the canyon and continental shelf transects. In this new section, we aim to integrate and reorganize information that was previously spread across different parts of the manuscript. Finally, we have also created section 3.5, entitled "Reanalysis data", which includes the use of the Mediterranean Sea Physics Reanalysis product (Escudier et al., 2020; 2021) to extend the temporal analyses of dense water transport beyond the observational period, and allows to place the winter 2021-2022 in the context of cascading events over the past two decades.

**Particular comments**

L51. What "it" makes reference to?
**Reply:** This was a typo. It referred to "these overflows" in the previous version. In the revised version, we have changed the sentence as follows (lines 55-59):

"As they descend, these overflows reshape the seabed by eroding and depositing sediments, and promote the downslope transport of organic matter accumulated on the shelf. Ultimately, DSWC in the GoL has been observed to impact biogeochemical cycles and the functioning of deep-sea ecosystems (Bourrin et al., 2006, 2008; Heussner et al., 2006; Sanchez-Vidal et al., 2008)".

L74-75. More prevalent than the extreme ones, thus, reducing overall DSWC over time? Or more prevalent than the "no DSWC scenario", thus, increasing overall DSWC over time?

**Reply:** Our sentence referred to the fact that, under climate change scenarios, MDSWC events are expected to become more frequent, while IDSWC events are projected to drastically decrease (in

occurrence and magnitude). We have revised the sentence in lines 85-87 as follows to make this statement clearer: "Although IDSWC events have drawn particular attention due to their significant impacts, mild DSWC (MDSWC) events are in fact the most frequent since the set of the observational era in the GoL, and they are expected to become more prevalent under the climate change scenario (Herrmann et al., 2008; Durrieu de Madron et al., 2023)".

During cold years, when IDSWC events occurs, most of the dense water formed over the shelf sinks into the deep ocean by deep cascading. In contrast, during warmer years associated with MDSWC events, the dense water is mainly consumed by mixing with lighter surrounding water, and only a small quantity escapes the shelf and produces shallow cascading. According to Herrmann et al. (2008), future projections indicate a significant reduction of dense water formation over the GoL's shelf, primarily due to the stronger stratification of the water column. This enhanced stratification results in a larger density gradient between surface and deep waters, making it more difficult for surface waters to break the stratification and reach deep layers. As a consequence, most of the dense water will be diluted through mixing, even in the coldest years, reducing the volume available to export. Thus, the fraction of dense water that effectively reaches the deep ocean through cascading will be much smaller in the future. Only a minimal amount is expected to escape the shelf, mainly flowing into the surface and intermediate layers (as MDSWC), leading to the disappearance of deep cascading (IDSWC).

L99-101. I'd remove: "which was monitored during the FARDWO-CCC1 cruise, and simultaneous measurements as its adjacent shelf acquired survey as part of the MELANGE-DUNES experiment" from here as it's too much detail for the introduction.

**Reply:** Done.

L118. Export of what? Just precise

**Reply:** Export of shelf water. We have clarified that in line 121 of the revised manuscript.

L129. What do you mean with "the concentration of water"? Are you referring to the residence time? Please rewrite, the term is awkward.

**Reply:** We agree that the sentence "the concentration of water" was unclear. We have rephrased the sentence and it now reads as: "where the continental shelf rapidly narrows and the Cap de Creus Peninsula constraints the circulation, intensifying the water flow and increasing the particle concentration (Durrieu de Madron et al., 1990; Canals et al., 2006)" (lines 130-133).

L.136. The full water column gets mixed? It would be surprising.

**Reply:** Thank you for your observation. We agree that the mixing of the full water column across the entire Gulf of Lion is unlikely. Our statement refers specifically to the continental shelf region. We have clarified this sentence in lines 139-141, which now reads: "The surface layers over the GoL shelf stratify between late spring and autumn (Millot, 1990). In winter, surface cooling and wind-driven mixing weaken the stratification, leading to a vertically homogeneous water column over the continental shelf (Durrieu de Madron and Panouse, 1996)."

L.149. 300-400 m is the upper limit I guess, above which stratification prevents the full mixing of the water column? In that case that would rather be a re-stratification, because DSW forms from the surface forcing, and then a light water layer develops in the surface. Is that it?

**Reply:** Thank you for your thoughtful comment. In this case, it is not a re-stratification process or the development of a lighter surface layer that limit the descent of DSW. During autumn or mild winters, DSW do not gain enough density when they are formed in the GoL to sink into the deep basin. Instead, they reach their equilibrium depth at intermediate depths, where they spread and contribute to the body of

Western Intermediate Water (WIW). We have revised the text accordingly to clarify this point (lines 150-158):

"During mild winters, these dense waters do not gain enough density (σ < 29.05 kg·m$^{-3}$) to sink into the deep basin, and contribute to the Western Intermediate Water (WIW) (T = 12.6-13.0 ºC and S = 38.1-38.3) body found at upper slope depths (~380-400 m) (Dufau-Julliand et al., 2004; Durrieu de Madron et al., 2005; Juza et al., 2013). The formation of WIW is an important process in the Mediterranean Thermohaline Circulation (MTHC), as it contributes to the ventilation of intermediate layers and plays a role in preconditioning the region for deeper convection events (Juza et al., 2019). During extreme winters, the potential density of DSW exceeds that of the EIW (σ = 29.05-29.10 kg·m$^{-3}$) and even surpasses the density of the WMDW (σ = 29.10–29.16 kg/m³), enabling DSWC to reach the deep basin around 2000-2500 m depth. This process contributes to the ventilation of the deep waters and to the final characteristics of the WMDW (Durrieu de Madron, 2013; Palanques and Puig, 2018)".

L.151. Gain

**Reply:** We have actually rephrased the sentence (155-157) as follows: "During extreme winters, the potential density of DSW exceeds that of the EIW (σ = 29.05-29.10 kg·m$^{-3}$) and even surpasses the density of the WMDW (σ = 29.10–29.16 kg·m$^{-3}$), enabling DSWC to reach the deep basin around 2000-2500 m depth".

L.164-165. However, all the point of TEOS10 is to promote the use of the more adequate conservative temperature and absolute salinity instead.

**Reply:** As we opted to use potential temperature and practical salinity to ensure consistency with previous studies and methodologies applied in the study area, we have removed the reference TEOS10 equation in the manuscript and deleted the corresponding references from the bibliography.

L.193-194. But what's the range of the bottom depth?

**Reply:** Thank you for your suggestion. We have now specified the bottom depth range along the glider section (83-92 m) in the revised version of the manuscript (lines 200-202).

L.216. Data is a plural noun: "Data were..."

**Reply:** Thank you. We have changed it throughout the revised version of the manuscript.

L.226-228. What type of data were used? Is it discharge volume?

**Reply:** The data correspond to river water discharge (expressed in liters per second) measured by gauging stations located near river mouths and provided by Hydro Portail v3.1.4.3 (https://hydro.eaufrance.fr). We have modified the text to add this information, which now stands as "Water discharge of rivers opening to the GoL was measured by gauging stations located near river mouths and provided by Hydro Portail v3.1.4.3 (https://hydro.eaufrance.fr)." in the revised version of the manuscript (lines 230-232).

L.286. Low compared to what? Give a reference please.

**Reply:** We agree that using the term "low" requires a reference or a baseline for comparison. To clarify our sentence, we have removed the word "low" (which is qualitative), and have rephrased the sentence as: "Significant wave height (Hs) ranged between 0.5 and 2.0 m (Fig. 3c) during winter" (329-332).

L.287. That's kind of surprising the existence of a storm that is not cause by strong winds, isn't it? Can you provide an explanation?

**Reply:** Thank you for your comment. We agree that labelling the storm as being caused by a "moderate" wind event might be confusing, given that Hs exceeded 3 m. To clarify, we have now specified in the

manuscript that wind speeds reached ~19 m·s⁻¹ and that the wind direction was easterly/southeasterly (E-SE) (lines 331-332). In the NW Mediterranean, E-SE winds are less frequent than the more dominant N-NW winds, but are typically associated with larger swells (Hs > 2 m and occasionally up to 10 m). They often occur simultaneously with river floods as the transport of humid marine air over the coastal promontory promotes heavy precipitation. The N-NW winds tend to produce only small waves (Hs < 2 m) over the inner shelf (Palanques et al., 2006).

The revised sentence in the manuscript states as: "Significant wave height (Hs) ranged between 0.5 and 2.0 m during winter (Fig. 3c). During this period, only one marine storm, defined as sustained Hs > 2 m for more than 6 hours (Mendoza and Jimenez, 2009), was recorded on March 13, 2022. This storm was associated with an easterly/south-easterly wind event with maximum speeds of ~19 m·s⁻¹, and generated Hs > 3 m for over 20 hours (Fig. 3c)".

L292. This is also surprising!

**Reply:** Following your observation, we carefully revised the time series of daily river discharge and the corresponding wind data (speed and direction). We realized that some wind directions were missing in the original plot. We have updated the figure accordingly, and we have found that the peak discharge over 5000 m³·s⁻¹ in late December was indeed associated with a brief easterly/southeasterly wind event. We have modified the text as follows: A peak discharge of over 5000 m³·s⁻¹ occurred in late December, associated with a brief easterly/south-easterly wind event (Fig. 3c)" (lines 335-336).

L.293. Low compared to which reference value?

**Reply:** Thank you for your question. We have now clarified this point by adding a reference to Bourrin et al. (2006), who provides daily average water discharges for the main coastal rivers discharging into the Gulf of Lion, including the Tech, Têt, Agly, Aude, Orb, Hérault, Lez, and Vidourle. We have included this citation in the text, and it now reads as "Coastal river discharges remained relatively low (see average daily water discharge values in Bourrin et al., 2006) during all the time period…" (lines 336-338), in order to provide context for what we considered "low" discharge.

Fig 3. It would be better to inverse the y-axis for density, so the densest water corresponds to the bottom layers.

**Reply:** Thank you for your comment. We agree with your suggestion, and have inverted the y-axis for density.

L319 and throughout the manuscript. It would be better to refer to the Moose stations by their location instead of LDC or CCC, which is complicated to remember.

**Reply:** As recommended, we change replaced the abbreviations "LDC" and "CCC" for the full names of the locations (Lacaze-Duthiers Canyon and Cap de Creus Canyon) throughout the manuscript to improve clarity. Additionally, we have slightly modified the text of this section as well as the caption of Figure 4 to clarify that LDC and CCC refer to Lacaze-Duthiers Canyon and Cap de Creus Canyon, respectively. We have retained the abbreviations in the figure.

L.336. Compared to what reference values? (please provide references whenever you state that XX values are low or high).

**Reply:** Noted. We have changed the text to avoid any confusion (lines 385-387).

Fig 5. Please avoid the used of divergent color maps for non-divergent fields as in the left column. This is misleading. Also, I'd personally prefer to see latitude instead of distance in the x-axis. I think it helps the readers to know where they are.

**Reply:** We agree with your comment. We have replaced the divergent colormap used for temperature in the left column of Fig. 5 for a non-divergent one, which we agree is more appropriate for representing this type of variable.

Moreover, we agree that using latitude can help the reader with geographic orientation. However, we have chosen to keep distance along the section on the x-axis because it is the most common approach in the literature, including the majority of the works cited in our manuscript. Additionally, in our case, the latitudinal variation along the section is relatively small, so we believe that replacing the distance with latitude will not substantially improve the interpretation of the figure. Nevertheless, the orientation and extent of the section is shown in Figure 1.

L341. This information belongs to methods. I actually missed it when I read it.

**Reply:** We agree with this comment. We have added this information to "Methods" (section 3.1.3, lines 200-202).

L.340-350. I suggest to better indicate what is from glider and what from cruise. It took me a moment to understand.

**Reply:** Thank you for your comment. We understand that the distinction between the glider-based data and cruise observations was not sufficiently clear at the beginning of this section. We have revised the first paragraph to explicitly indicate that it refers to glider data. Also, we have added a transition sentence at the end of the paragraph and at the beginning of the second one to specify that the T1 and T2 transects were conducted during the FARDWO-CCC1 Cruise. We hope these changes make it easier to follow the different observations in the Cap de Creus Canyon and the continental shelf.

We have also renamed the transects in Figures 5, 6, and 7 by location, which now are "Continental shelf (glider transect)", "Upper canyon (T1 transect)", and Mid canyon (T2 transect)".

Fig 6: The color bars for panels f and i are not the same, even if they have the same limits and correspond to the same variables, which is misleading and makes comparison difficult.

**Reply:** We have carefully reviewed the figure. We have replaced the previous colour scale for a continuous one and ensured that panels **(f and i)** share the same limits and colour mapping. Moreover, we have updated the colour scale in the glider transect (Fig. 6c), although it has a different range to better visualize the oxygen values in the continental shelf.

L430. However, the discharge was low this winter, and dense water forms other years. This makes me think that this is not a reason to justify the low density.
**Reply:** We agree that the way the text was written may suggest that river discharge was the main reason for the density gained by shelf waters. In fact, the density gained by shelf waters depends mostly on the atmospheric forcings (heat losses). Freshwater inputs from the Rhône River and the coastal rivers contribute to localized freshening. We have rewritten this section (5.1.) to make this statement clearer.

L.432-435. I can't really see a decrease in density, which makes me think that river discharge is not a key factor.
**Reply:** You are right. The higher discharge of the Rhône River and coastal rivers during winter 2021-2022 does not show a direct or clear link to a decrease in shelf water density. We have revised Section 5.1. to better reflect this point. In fact, the density of shelf waters reached 28.9 kg·m$^{-3}$, which was insufficient to overcome the Eastern Intermediate Water (EIW) layer and trigger deep cascading. Instead, this MSWC event likely contributed to the body of Western Intermediate Water (WIW), as described in previous studies (Dufau-Julliand et al., 2004).

Fig 8. Wouldn't it be better to plot bottom density in order to identify dense water? Also, please change the color map for a non-divergent one. This one is misleading.

**Reply:** We agree with your suggestion. The figure now shows the bottom density to better identify dense shelf waters over the continental shelf. Also, we have also replaced the previous colormap with a non-divergent one.

L.446-447. As I said above, we cannot judge if the values are low or high if we don't have references.
**Reply:** Agree. We now mention reference values instead of "low/high" and include a comparison with previously reported IDSWC events (Canals et al., 2006) at the end of Section 5.1 (lines 498-500).

L479. Suggest.
**Reply:** Changed.

L489. Flows.
**Reply:** Changed.

L.500-510. This paragraph should definitely go to Methods and not in the discussion.

**Reply:** As suggested, we have moved this information to a new dedicated section in Methods ("3.4. Estimation of dense water and SPM transports from observations").

L.513. 0.05 Sv is practically zero, taking into account the strong variability. I actually would say the mean is negative? Have the authors double checked this mean? In any case, given the difference in the T1 and T2 value, I would not define the Cap de Creus Canyon as a partial sink, it is rather not at sink during mild winters. Whether or not this canyon is a sink, or export occurs through it remains confusing to me throughout the manuscript.

**Reply:** We have double checked our calculations and confirm that they are correct, even if the resulting transport is low. However, we acknowledge that referring to the Cap de Creus Canyon as a "partial sink" may have caused a bit of confusion. Our point was to highlight that during mild winters, such as the presented in our paper, the canyon still acts as a conduit for dense shelf waters, but only to a limited extent (upper canyon), in contrast to extreme winters. We have removed this term throughout the manuscript and revised the text to emphasize that cascading was mainly confined to the upper canyon, with weaker signals reaching the mid-canyon section. We hope that this interpretation is now clearer.

L519-520. You state you used the reanalysis "to assess the variability of dense shelf water export in the Cap de Creus Canyon during the mild winter of 2021-2022." but the computation spans the October-May period, so, beyond winter.

**Reply:** You are correct that the original analysis expanded beyond the winter season. We have revised the figure and changed the timeframe to include only the winter months (December, January, February, and March), which are the most relevant for the occurrence of cascading events. The manuscript has also been updated to reflect this change in section 5.3.2.

L.525. I miss having some numbers to compare the reanalysis with the observations and quantify how well they match. You should plot the same variable for the T1 and T2 transects, integrated over the same depths. You could event add a line for the value of each variable in your observations. This would provide robustness to the reanalysis results.

**Reply:** It is not possible to add a line for each of our observation values on the reanalysis time series of Figure 9 because our observations are based on data from CTD casts obtained on a specific day (a snapshot). Therefore, we cannot provide this comparison on a time series.

Nevertheless, we carried out a comparison between our observations and reanalysis data. First, we analysed all stations by filtering those that met the dense water temperature criteria (T < 12.9 ºC). For

these stations, we calculated the depth-averaged temperature within the range occupied by the dense waters. We applied the same procedure to the reanalysis data over the corresponding locations and time. Finally, we compared the resulting depth-averaged temperatures from observations and reanalysis using the root mean square method (RMSE) statistical method, which allows to estimate the deviation (or residuals) of the predicted values (reanalysis) from the observations (Table 1). In general, RMSE are below 0.2 ºC, which shows a good agreement between both datasets, and supports the reliability of using this reanalysis product in our study to assess the temporal variability of dense water transports.

Moreover, as previously commented, we have recently submitted a paper to the same journal (Fos et al., 2025) in which we conduct a thorough statistical analysis and validate this reanalysis product against long-term mooring observations in the Cap de Creus and Lacaze-Duthiers canyons. In that paper, we demonstrate that reanalysis accurately reproduces DSWC events, matching 84% of IDSWC events within the same week and 56% on the exact date. This validation further reinforces the robustness and applicability of the reanalysis data that we use in our paper.

L.546. "relatively weak wind forcing".

**Reply:** Noted.

L.560-562. How was this percentage estimated? I'm a bit confused. When we say export, I think about the water transport down-canyon to reach deeper depths, if water doesn't get to leave the shelf I wouldn't call it export. Throughout the manuscript the authors state (and the transport numbers suggest) that the actual export is very weak. I would like to know how these percentage were computed and, as asked before, what are the reference values in Sv (for instance a climatological mean, or the typical values in strong winters) for transport.

**Reply:** We have removed the reference to percentages, as we did not explicitly calculate the portion of dense waters that flowed along the coast versus the portion that was actually transported through the canyon. We agree that including these percentages without a clear reference is misleading.

On the other hand, we have decided to retain the term "export" when referring to the downslope transport of dense shelf waters into the canyon. We think it is an appropriate term since there was indeed a net downslope transport of dense shelf water, although with a much lower magnitude than in extreme winters.

Finally, to better contextualize the weak export in winter 2021-2022, we have added a reference to Fos et al. (2025), which reports a peak in dense water transport of 1.29 Sv in the Cap de Creus Canyon for the IDSWC event of winter 2004-2005. Additionally, we now include comparisons between the estimated exported volumes (in km³) with those reported for other mild and extreme winters, in order to provide a clearer view of the interannual variability of dense water export through the canyon (lines 631-639). In this context, we have also incorporated a new figure (Figure 10) with a long time series (from 1997 to 2022) better contextualize and compare our cascading event (2021-2022) with previously reported events.

**References:**

[revised manuscript text omitted]

---

## Author Comment (AC2)

**Response to reviewer 1**

Dear reviewer,

We appreciate your constructive and relevant comments and suggestions to our manuscript. Below, your reviews are reproduced in **black**, while our comments are in **blue**.

Since the reviewer #2 also raised important points, we have made substantial changes throughout the manuscript. Some of these changes may also address your comments or provide useful context for the paper, so we kindly suggest to take a look at our responses to Reviewer #2 as well.

Please, note that all **line numbers** in our responses refer to the clean version of the manuscript, not the tracked-changes version.

**"Dense shelf-water and associated sediment transport in the Cap de Creus Canyon and adjacent shelf under mild winter regimes: insights from the 2021–2022 winter" by Arjona-Camas et al.**

**General Comments:**

This manuscript presents a well-written and carefully conducted observational study of dense shelf water cascading (DSWC) and associated sediment transport in the Cap de Creus Canyon during a mild winter (2021–2022). The authors employ a multi-platform approach—including gliders, moorings, ship-based CTD profiles, and reanalysis data—to describe the cascading evolution and to estimate transport of water masses and suspended sediments.

The manuscript is well structured and clearly written, with high-quality figures and solid data processing. However, the conceptual novelty is limited, as the key findings align closely with what is already established in the DSWC literature. Specifically, prior studies—including Mahjabin et al. (2019, Continental Shelf Research; 2019, JMSE; 2020, Scientific Reports)—have demonstrated:

- That DSWC can occur under mild to moderate wind forcing;

- That wind direction is a key modulator of cascading strength;

- That such events result in substantial sediment and biogeochemical transport.

Moreover, these studies introduced predictive frameworks such as the Simpson number and energy balance models, and examined canyon-free shelf settings under similar climatic regimes. These works are not cited in the current manuscript.

While the present study is geographically focused on the Cap de Creus Canyon, the manuscript could benefit from a deeper exploration of canyon-specific dynamics—such as flow steering, internal hydraulics, or sediment redistribution mechanisms—which are only briefly mentioned. Additionally, while the observations are carefully described, the broader significance of this mild-winter case for global DSWC understanding is not yet fully articulated. A more explicit discussion of the study's unique contribution—particularly in terms of sediment asymmetry, constrained cascade depth, and implications for WIW formation—would significantly enhance the manuscript's impact.

**Reply:** We appreciate this overall positive assessment of our manuscript, and we thank you for pointing out both its strengths and the areas for improvement.

We agree that the Introduction would benefit from citing other studies of DSWC in different settings. We have added a paragraph in the Introduction (lines 34-43) about dense shelf water cascading, as well as its implications in the global ocean. We have also added references to documented cases of DSWC around the world, including the studies you recommended on cascading off Australia (Mahjabin et al., 2019, 2020).

We acknowledge that concepts such as flow steering, internal hydraulics, and sediment redistribution mechanisms are important canyon-specific processes. Our data do not allow for a full dynamical analysis of these mechanisms. However, we have expanded the discussion of our paper by including:

- Lines 518-536: Determination of the Richardson (Ri) and Froude numbers (Fr), which provide insights into the stability and dynamical behavior of stratified flows. Ri values showed a general increase between 150 and 300 m depth, which roughly corresponds to the vertical extent of the dense shelf water plume. The maximum Ri observed reached 0.18 at 270 m depth in the upper canyon, and 0.16 at 180 m in the mid canyon. These values are below the critical threshold of 1 that separates laminar (Ri >1) from turbulent flow regimes, thus indicating a predominantly turbulent flow (Mack and Schoeberlein, 2004). This suggests that fluid instabilities likely enhanced vertical mixing and lateral spreading of the dense water plume. Additionally, we obtained Fr ~1.10. This value lies just above the critical threshold of 1, indicating a supercritical flow regime where inertial forces become more significant, potentially favoring more unsteady and turbulent flow conditions (Cenedese et al., 2004).

- Lines 552-569 and lines 570-580: Discussion on how the geomorphology of the canyon influences the redistribution of sediments in the canyon. In addition, we have added lines 718-721 to acknowledge that future research would benefit for an in-depth analysis of the physical dynamics that drive DSWC. Thank you again for these constructive suggestions.

**Specific comments:**

**1) On novelty and contextualization**
The Gulf of Lions is among the most studied regions globally for DSWC, with numerous works documenting both mild and extreme cascading events. While the present manuscript focuses on a specific mild winter (2021–2022), the authors should more clearly state what new understanding this adds. For example: Is the sediment asymmetry across the canyon novel? Is the observed upper canyon confinement unusual for mild winters? More detailed differentiation from earlier work is encouraged.

**Reply:** We appreciate your suggestion to better state the relevance of our study. For that, we have rewritten the Introduction and added some lines with the knowledge gap and enhance the novelty of our study.

There are several studies conducted in the GoL investigating both intense dense shelf water cascading (IDSWC) events -such as Heussner et al. (2006), Canals et al. (2006), or Durrieu de Madron et al. (2013)- and mild DSWC (MDSWC) in the Cap de Creus Canyon -such as Ulses et al. (2008a), Martín et al. (2013), Rumín-Caparrós et al. (2013), or Mikolajczak et al. (2020)-. These studies are mostly based on mooring time series acquired in the canyon head and/or model outputs and numerical simulations to detect the presence of dense waters and infer their export pathways, which offer limited spatial resolution and lack direct observations on shelf-slope transports. To our knowledge, there has been no comprehensive observational characterization of dense water and sediment transport from the shelf to the slope in the Cap de Creus Canyon under moderate winter conditions during MDSWC events. To address this gap, we offer a combination of hydrographic and velocity measurements collected concurrent within the canyon and the adjacent shelf to resolve the shelf-to-slope transport of dense waters and associated suspended sediment, along with reanalysis data to determine the temporal extent of the 2021-2022 MDSWC event and place it in the context of cascading events observed in the Gulf of Lion over the last 26 years. This latter part is new, but we believe it will help us to strengthen the importance of our work and generalize our conclusions.

**2) Wind Direction and Episodic Forcing**
The manuscript appropriately links SE wind events to episodic downwelling and DSWC initiation. However, this connection is largely descriptive. Including wind stress time series or Ekman transport estimates would strengthen the argument and provide a more quantitative link to the observed cascading pulses.

**Reply:** We appreciate your suggestion and fully agree that a quantitative analysis of different forcings, such as wind stress time series or Ekman transport estimates, could provide a more quantitative characterization of the observed DSWC. However, this goes beyond the scope of the present study, which does not aim to investigate the physical dynamics of DSWC or the processes driving the initiation and evolution of cascading in detail. However, and in agreement with previous studies, we can still infer aspects on the dynamics of the dense water plume based on existing theoretical frameworks. For example, if we schematize the behavior of the cascading plume observed in our study for winter 2021-2022 using the classification of Shapiro and Hill (2003) (which describe the effect of friction on dense water plumes), our observations suggest a "head-up" configuration. This means that most of the dense fluid remains upslope, while only a thin tail drains downslope. Also, the steepest isopycnals occur on the upslope (western) side, while the downslope side (toward the shelf break) remains thinner. We have added a line discussing this interpretation in the Discussion section (lines 518-522). As previously stated, we have also estimated the Richardson and Froude numbers to determine the flow regime of the plume. We now discuss it in lines 518-536.

That said, we acknowledge the importance of conducting a more quantitative analysis of the different forcings involved, and we agree that future work would benefit from this detailed exploration of the physical dynamics of DSWC (lines 718-721).

**3) Canyon-Specific Dynamics**
While the Cap de Creus Canyon is central to the title and framing, the manuscript does not deeply examine its dynamic role beyond being a conduit. Consider discussing whether canyon morphology contributes to observed sediment asymmetries or restricts flow depth. Alternatively, consider softening the canyon emphasis if the goal is to document a shelf-wide mild DSWC event.

**Reply:** We acknowledge that canyon morphology contributes to the sediment asymmetries observed in our transects. We have expanded the discussion to include a more detailed explanation on how the canyon's morphology may have influenced the increased SPM concentrations associated with dense waters on the southern canyon flank (see section 5.2.)

**4) Citation Inclusion**
Please cite the following prior studies if relevant:

- Mahjabin, T., Pattiaratchi, C., & Hetzel, Y. (2019a). *Wind effects on dense shelf water cascades in south-west Australia.* Continental Shelf Research, 189, 103975.

- Mahjabin, T., Hetzel, Y., & Pattiaratchi, C. (2019b). *Spatial and temporal variability of dense shelf water cascades along the Rottnest continental shelf in southwest Australia.* JMSE, 7(1), 30.

- Mahjabin, T., Pattiaratchi, C., & Hetzel, Y. (2020). *Dense shelf water cascading around the Australian continent.* Scientific Reports, 10, 9930.

These studies support the notion that DSWCs can occur under non-extreme conditions and offer theoretical and methodological insights that are directly relevant here.

**Reply:** Thank you for the suggestion. We agree that these studies are relevant to our work, as they provide insights into DSWC under non-extreme conditions in other continental margins. As recommended, we have now cited Mahjabin et al. (2019b; 2020) in the Introduction to reinforce the broader context in which DSWC occurs across diverse continental margins and latitudes.

**Technical Corrections**
**Abstract:** The opening sentence *"This study examines…"* is generic and does not effectively convey the study's context or significance. I recommend replacing it with a more engaging and informative sentence that introduces DSWC and the knowledge gap being addressed. For example: *"Dense shelf water cascading (DSWC) is a key process in transferring water masses and sediments from*

*continental shelves to deep basins, yet its dynamics under mild winter regimes remain poorly characterized."*

**Reply:** We have now added this informative sentence in the abstract (lines 9-10).

**Introduction:** While DSWC is mentioned early, it is not clearly defined. I recommend including a short, reader-friendly definition in the introduction, such as: *"DSWC refers to the downslope flow of cold, dense water formed on continental shelves due to surface cooling and/or evaporation, which descends under gravity into deeper ocean basins."*

**Reply:** Thank you for pointing this out. We agree that a clearer introduction to the process of dense shelf water cascading was necessary to improve clarity for the reader. Following your suggestion, we have included a new paragraph in the Introduction (lines 34-43) describing DSWC and its global implications.

Line 236: Typo — "metter" should be corrected to "meter".

**Reply:** Changed.

**References:**

Canals, M., Puig, P., Durrieu de Madron, X., Heussner, S., Palanques, A., and Fabres, J.: Flushing submarine canyons. Nature 444, 354-357, https://doi.org/10.1038/nature05271, 2006.

Cenedese, C., Whitehead, J. A., Ascarelli, T. A., and Ohiwa, M.: A dense water current flowing down a sloping bottom in a rotating fluid. J. Phys. Oceanogr. 34, 188-203, https://doi.org/10.1175/1520-0485(2004)034%3C0188:ADCFDA%3E2.0.CO;2, 2004.

Durrieu de Madron, X., Houpert, L., Puig, P., Sanchez-Vidal, A., Testor, P., Bosse, A., Estournel, C., Somot, S., Bourrin, F., Bouin, M. N., Beauverger, M., Beguery, L., Canals, M., Cassou, C., Coppola, L., Dausse, F., D'Ortenzio, F., Font, J., Heussner, S., Kunesch, S., Lefevre, D., Le Goff, H., Martín, J., Mortier, L., Palanques, A., and Raimbault, P.: Interaction of dense shelf water cascading and open-sea convection in the northwestern Mediterranean during winter 2012. Geophys. Res. Lett. 40, 1379-1385, https://doi.org/10.1002/grl.50331, 2013.

Heussner, S., Durrieu de Madron, X., Calafat, A., Canals, M., Carbonne, J., Desault, N., and Saragoni, G.: Spatial and temporal variability of downward particle fluxes on a continental slope: lessons from an 8-yr experiment in the Gulf of Lions (NW Mediterranean). Mar. Geol. 234, 63-92, https://doi.org/10.1016/j.margeo.2006.09.003, 2006.

Mack, S. A., and Schoeberlein, H. C. : Richardson number and ocean mixing : towed chain observations. J. Phys. Oceanogr. 34, 736-754, https://doi.org/10.1175/1520-0485(2004)034<0736:RNAOMT>2.0.CO;2, 2004.

Mahjabin, T., Pattiaratchi, C., Hetzel, Y., and Janekovic, I.: Spatial and temporal variability of dense shelf water cascades along the Rottnest continental shelf in southwest Australia. J. Mar. Sci. Eng. 7, 30, https://doi.org/10.3390/jmse7020030, 2019.

Mahjabin, T., Pattiaratchi, C., and Hetzel, Y.: Occurrence and seasonal variability of Dense Shelf Water Cascades along Australian continental shelves. Sci. Rep. 10, 9732, https://doi.org/10.1038/s41598-020-66711-5, 2020.

Martín, J., Durrieu de Madron, X., Puig, P., Bourrin, F., Palanques, A., Houpert, L., Higueras, M., Sánchez-Vidal, A., Calafat, A. M., Canals, M., Heussner, S., Delsaut, N., and Sotin, C. : Sediment transport along the Cap de Creus canyon flank during a mild, wet winter. Biogeosciences 10(5):3221-3239, https://doi.org/10.5194/bg-10-3221-2013, 2013.

Mikolajczak, G., Estournel, C., Ulses, C., Marsaleix, P., Bourrin, F., Martín, J., Pairaud, I., Puig, P., Leredde, Y., Many, G., Seyfried, L., and Durrieu de Madron, X.: Impact of storms on residence times and export of coastal waters during a mild autumn/winter period in the Gulf of Lion. Cont. Shelf Res. 207, 104192, https://doi.org/10.1016/j.csr.2020.104192, 2020.

Rumín-Caparrós, A., Sanchez-Vidal, A., Calafat, A. M., Canals, M., Martín, J., Puig, P., and Pedrosa-Pàmies, R.: External forcings, oceanographic processes and particle flux dynamics in Cap de Creus submarine canyon, NW Mediterranean. Biogeosciences Discussions 9(12), https://digital.csic.es/handle/10261/78113, 2013.

Shapiro, G. I., and Hill, A. E. : The alternative density structures of cold/saltwater pools on a sloping bottom : the role of friction. J. Phys. Oceanogr. 33, 390-406, https://doi.org/10.1175/1520-0485(2003)033<0390:TADSOC>2.0.CO;2, 2003.

Ulses, C., Estournel, C., Bonnin, J., Durrieu de Madron, X., and Marsaleix, P. : Impact of storms and dense water cascading on shelf-slope exchanges in the Gulf of Lion (NW Mediterranean). J. Geophys. Res. Oceans, 113, http://doi.org/10.1029/2006JC003795, 2008a.

---

## Referee Report (RR1)

**2nd revision of the Manuscript by Arjona-Camas et al. by Esther Portela**

I appreciated the detailed response of the authors to my comments. I particularly liked the inclusion of a broader context by using the reanalysis data. However, I think the last figure should be better discussed, while other parts of the paper seem much less relevant to me and could be streamlined. I find the paper very long with a too detailed results description and with part of the discussion that really doesn't belong there. Discussion seems to be much longer than in the previous version and there are entire parts that don't seem relevant to me (as I stated in my comments below), but I could be missing something important, and if that's the case, please let me know.

My main criticism is still the same, that despite the nice data compilation, I still think this study doesn't add much and has little implications regarding what is already known. That said, I am a big fan of exploiting all available data to address different scientific questions before new data collection with the associated carbon print. I also don't think research has to be always innovative, but interesting and well conducted, which is largely the case here. Because of that, and since I'm not an expert of this region, I prefer not to make a strong judgment on the novelty of this study, and will just trust the other reviewer's and editor opinion about this point.

**Please find my comments below**

L20-23: I still find this sentence a bit ambiguous regarding the magnitude and extent of the DSW cascading and export. Mainly it is the term "export" what bothers me. What does "export" refer to? Can we use this term when we find the given water mass to be transported 100 km away? 10 km away? 1 km away?

L47: This is what I understand by "export"

**Methods.**

I appreciate the authors gave us more information about the interpolation (or gridding) method. But I still think more details are needed, mostly about the spacing of the grid or the length scales used (if so)

L.234. SPM has not been defined at this point

**Results. Section 4.1.**

I find a bit hard to interpret this section if I still don't know what, how and when, is related to the dense water export.

L361-363. Actually the strongest ocean heat loss starts around November 1st and goes until approx the end of February.

L334-335. Why should discharge be related to the wind?

L336-340. And related to the Hs, isn't it?

**Main comment:**

L397-398. I feel like this sentence should be the beginning of the story. The authors chose to provide the context before showing the presence of Dense Shelf Water, which is the object of the study. As I said in my previous comment, all that previous information is kind of empty if

we don't know where is the DSW observation and how it looks like. It is a matter of style maybe, but I'd find it much clearer the other way around.

Fig 4e. Why is the y-axis scale so large? You could reduce them a lot so the variability would be much more visible. Currents seem to be nearly zero at the CCC during the cruise time, which is actually the focus of this study. I also wonder what does the alternating positive negative current pattern means. Do you have an explanation for the up-canyon and down-canyon currents to be so regular?

Fig 5. Please make the axes labels larger.

L406. I can't see the high dissolved oxygen as compared with the upper layer.

L461-463. Try to avoid subjective language, mostly when numbers are provided, and let the reader decide. 150 W m-2 is 25% less than 200 W m-2, and is half of 500 W m-2.

Discussion: I find the discussion to be way too long. While the writing is very clear and the connection with the bibliography is excellent, the new discussion is substantially longer than in the previous version, with many additions for which I do not really see the relevance (unless it's in response of the reviewer comments, but even then..). There are also lots of repetitions of the results numbers and references to figures. I would recommend to summarize and streamline, I've made more concrete suggestions about this in the following comments.

L495-496. But as important as talking about current speed is current direction.

L519-535. I cannot see the point of these lines. This part seems disconnected from the rest of the study, it does not really provide any useful information, and is also misplaced in the discussion section.

L551-169. I find this paragraph too long. Mostly because it only addresses a small part of the results, the SPM concentrations in the plume, which is also not so surprising.

L582-583. This is an example of results repetition than can be avoided.

L591- From here, I'd say this belong to results and not to the discussion section. Also, the comparison would be more clearer with a map (even in the S.I), as with the data provided in the table we can't really see how well the spatial patterns are represented. For instance, the reanalysis provides 12.42°C for T1-03, and T104, while their observed mean temperature differed by 0.4°C.

L605-606 I guess this means that the transport has been averaged for the dense waters density layers. But then, There are many instances with up-canyon transport, which is quite weird, isn't it? Or is there an explanation for this?

Figure 10 is nice, but I miss more discussion around it. Also can you explain how did you differentiated between mild and intense events? It this a wind, or heat loss threshold as you mentioned in the introduction? It is surprising that transport is often as intense in mild events as in intense events.

---

## Referee Report (RR2)

**Review Comments:**

The authors have substantially improved the manuscript in response to the previous round of comments. The study presents a well-structured observational analysis of dense shelf water cascading (DSWC) and sediment transport in the Cap de Creus Canyon during a mild winter regime. The multi-platform dataset (moorings, gliders, CTD profiles, and reanalysis products) is robust, and the results provide useful insights into the dynamics, timing, and sediment export processes under mild winter conditions.

**Novelty Assessment:**

The novelty is somewhat limited because DSWC in the Cap de Creus Canyon under mild winter conditions has previously been described by *Martin et al.* (2013) for the 2010–2011 winter, including estimates of dense water transport (~0.3 Sv) and sediment load (~105 t). The present study adds:

- A more recent mild-winter case (2021–2022) with higher-resolution, multiplatform observations.
- Measurements across both the continental shelf and canyon transects.
- Integration of hydrodynamic and sediment transport data with updated reanalysis products.

At present, the novelty is primarily methodological and contextual rather than conceptual. However, it can be strengthened by including an explicit quantitative comparison of transport and sediment load values between:

- 1. The present mild winter (2021–2022),
- 2. The previous mild winter (2010–2011; Martín et al., 2013), and
- 3. Known strong-winter events (e.g., Canals et al., 2006; Puig et al., 2008).

Such a comparison would position the study as the first to place recent mild-winter dynamics into the broader spectrum of DSWC intensities in the Cap de Creus Canyon, increasing its interpretive value and relevance for understanding climate-driven variability in cascading processes.

**Abstract Clarity:**

The sentence "...yet its dynamics under mild winter regimes remain poorly characterized" should be qualified to avoid implying a global knowledge gap. Since mild-winter DSWC has been documented elsewhere (e.g., Mahjabin et al., 2019, 2020), and even in the Cap de Creus Canyon (Martín et al., 2013), I recommend revising to this line.

for example it can be written as:

"...yet its dynamics under mild winter regimes in the northwestern Mediterranean, particularly in the Cap de Creus Canyon, have been less comprehensively described and compared to strong-winter events."

**This way:**

- It narrows the scope to **region + site** (avoids implying a global knowledge gap).
- It acknowledges some existing work (e.g., Martín et al. 2013) but still
  justifies the new study.
- It sets up the importance of **comparison with strong winters** early in the paper.

**Minor Corrections and Consistency Edits:**

- SI unit for metric tonnes Use the correct SI symbol: t (lowercase). At first occurrence, write as t (metric tonnes), and thereafter use t alone. Ensure a space between the value and the unit (e.g., "105 t", not "105t"). Replace non-SI or ambiguous forms such as "metric tons" or "T" where applicable.
- **Hyphenation** Standardize usage to either *dense shelf water cascading* (no hyphen) or *dense shelf-water cascading* (with hyphen) throughout text and captions.
- **Acronyms** In Section 3.2.1, correct ECMWF to *European Centre for Medium-range Weather Forecasts* (ECMWF).

**Overall Recommendation:**

With these relatively minor edits and an expanded discussion comparing the present results with both previous mild-winter and strong-winter events, the manuscript will be well-prepared for publication in Ocean Science. The observational dataset is valuable, the analyses are sound, and the study adds meaningful insight into DSWC dynamics in a mild winter regime for this specific canyon system.

---

## Referee Report (RR3)

**Review Comments**

The authors have substantially improved the manuscript in response to previous reviewer comments. The study presents a solid observational analysis of dense shelf water cascading (DSWC) and sediment transport in the Cap de Creus Canyon during the mild winter of 2021–2022. The dataset is comprehensive (moorings, gliders, CTD profiles, and reanalysis products), and the comparisons with both previous mild-winter and strong-winter events now place the results in the correct broader context. The manuscript is much clearer and better framed than earlier versions.

**Remaining Issues**

**1. Table 1**

Table 1 is useful for transparency (particularly for RMSE validation values and comparison with *Martín et al., 2013*), but the essential findings (RMSE consistently <0.2 °C; quantitative comparability across events) are already described in the text. To streamline the manuscript, I recommend moving Table 1 to the Supplementary Material and retaining only a summary statement in the main text.

**2. Minor corrections**

- **Acronyms** In Section 3.2.1, correct ECMWF to *European Centre for Medium-range Weather Forecasts (ECMWF)*.
- **Hyphenation** Standardise usage of *dense shelf water cascading* (no hyphen) or *dense shelf-water cascading* (with hyphen) consistently across text and figure captions. So to be consistent just remove the hyphen from Title.

**Overall Recommendation**

With these final minor corrections, the manuscript will be well-prepared for publication in *Ocean Science*. The observational dataset is valuable, the analyses are sound, and the paper now provides meaningful and well-contextualised insights into DSWC dynamics in a mild winter regime for the Cap de Creus Canyon.

---

## Author Response (AR2)

**Response to Editor**

Dear Editor,

We appreciate the constructive feedback provided by the reviewers.

Reviewer #1 raised concerns regarding the novelty of our study, particularly the potential overlap with Martín et al. (2013), as well as the overall balance of the manuscript sections. To address these points, we have revised the text to streamline the Results and Discussion sections, avoiding repetitions, and reducing descriptive details. The new Discussion follows a more logical order, better highlights the results of the paper, and places them in context by comparing the winter of 2021-2022 with previous mild and intense cascading events.

To clarify the novelty of our manuscript, and remove any ambiguity about the novelty of our work, we have prepared a comparative table (also included in our response letter to Reviewer #1) that clearly demonstrates the substantial advances and unique contributions of our study beyond Martín et al. (2013).

**Comparative table: our study vs. Martin et al. (2013)**

| Aspect                         | Martín et al. (2013)                                                                                                                                                               | This study                                                                                                                                                                                                                                                       |
|--------------------------------|------------------------------------------------------------------------------------------------------------------------------------------------------------------------------------|------------------------------------------------------------------------------------------------------------------------------------------------------------------------------------------------------------------------------------------------------------------|
| Process investigated           | Downwelling of coastal waters                                                                                                                                                      | Mild/shallow DSWC event                                                                                                                                                                                                                                          |
| Period studied                 | March 2011                                                                                                                                                                         | Winter-spring 2021-2022 (with detailed observations in March 2022)                                                                                                                                                                                               |
| Instrumentation                | Moorings + CTD transect at the southern canyon flank                                                                                                                               | Multi-platform approach: moorings + hydrographic transects (glider + ship-based CTD) at the shelf and canyon + ADCP + reanalysis product.  Original observations and new data collected during the FARDWO-CCC1 Cruise (2022).                                    |
| Main forcings                  | Storm-induced downwelling                                                                                                                                                          | Moderate net heat losses during winter. Eastern storms further enhanced DSWC                                                                                                                                                                                     |
| Water mass characteristics     | Pot. temp.~ 11.5-12.5 °C
Salinity ~ 37.7
Pot. density ~ 28.78 kg·m -3                                                                                             | Pot. temp.~ 12.2-12.7 o C
Salinity ~ 38.1-38.2
Pot. density ~ 28.9-29.1 kg·m -3                                                                                                                                                      |
| Dense water volume             | No significant dense shelf water formation.  Downwelling to the canyon head ~0.2 Sv and 10 5 t of SPM                                                                   | Dense shelf water formed in the Gulf of Lion during winter 2021-2022.  0.7 Sv and 10 5 t SPM cont. shelf  0.3 Sv and 10 5 t SPM upper canyon  0.05 Sv and 10 4 t SPM mid canyon                                                 |
| Detachment depths              | 200-300 m depth                                                                                                                                                                    | 150-400 m depth                                                                                                                                                                                                                                                  |
| Main contribution of the paper | Documentation of a storm-induced downwelling event, with the absence of DSWC under limited external forcing during a mild winter.  No comparison with previous events is provided. | Detailed observational characterization of a mild/shallow DSWC event, as well as the shelf-slope dense-water and sediment transport during a mild winter. Comparison with previous mild and extreme events (broader spectrum of cascading events in the region). |

We believe that all these elements demonstrate that our manuscript presents original observations of a shallow/mild cascading event, extending previous findings in the canyon under similar meteorological regimes, and contributing to a better understanding of the interannual variability of both mild and intense DSWC events in the Cap de Creus Canyon. We have made the pertinent changes throughout the manuscript to highlight these differences. We hope these revisions have addressed her concerns about length, focus, and balance.

Reviewer #2 mainly suggested minor clarifications, which we have incorporated to the manuscript. Also, she suggested to include a comparison between this study with previous mild and intense cascading events. Accordingly, we have expanded the Discussion section to include it. Our comparison considers the atmospheric forcings and dense water transport values, which allows us to put our study within the broader spectrum of cascading intensities in the Cap de Creus Canyon.

**Response to reviewer 1 (2nd revision)**

**Dear reviewer,**

We thank you very much for your relevant comments to our manuscript. Below, your reviews are reproduced in **black** font and our responses in **blue**. Since the other reviewer has raised important points, we kindly suggest to review her responses. Please, note that all line numbers in our responses refer to the **clean version** of the manuscript, not the tracked-changes version.

**2nd revision of the Manuscript by Arjona-Camas et al. by Esther Portela**

I appreciated the detailed response of the authors to my comments. I particularly liked the inclusion of a broader context by using the reanalysis data. However, I think the last figure should be better discussed, while other parts of the paper seem much less relevant to me and could be streamlined.

I find the paper very long with a too detailed results description and with part of the discussion that really doesn't belong there. Discussion seems to be much longer than in the previous version and there are entire parts that don't seem relevant to me (as I stated in my comments below), but I could be missing something important, and if that's the case, please let me know.

My main criticism is still the same, that despite the nice data compilation, I still think this study doesn't add much and has little implications regarding what is already known. That said, I am a big fan of exploiting all available data to address different scientific questions before new data collection with the associated carbon print. I also don't think research has to be always innovative, but interesting and well conducted, which is largely the case here. Because of that, and since I'm not an expert of this region, I prefer not to make a strong judgment on the novelty of this study, and will just trust the other reviewer's and editor opinion about this point.

**Reply:** We thank you for your constructive comments. We have carefully considered your suggestions and made substantial revisions to improve the clarity, focus, and balance of the paper.

*Last figure:* We have added more discussion around Figure 10 (lines 524-536), comparing the winter 2021-2022 with previous mild and strong winters in terms of transports and associated forcing conditions.

**Streamlining the Results and Discussion sections:** We have carefully revised the Discussion section to avoid repetitions with the Results, removed information that was less connected or added limited discussion to our paper, and incorporated additional discussion. We believe that this final version of the Discussion (from line 504) follows a more logical order, better highlights the results of the paper, and places them in context by comparing them (with a bit more detail) with previous mild and intense cascading events.

**Novelty and contribution of our study:** To remove any ambiguity regarding the novelty of our work, we have prepared a comparative table that clearly demonstrates the substantial advances and unique contributions of our study beyond Martín et al. (2013).

**Comparative table: our study vs. Martin et al. (2013)**

| Aspect               | Martín et al. (2013)                                 | This study                                                                                                                                |
|----------------------|------------------------------------------------------|-------------------------------------------------------------------------------------------------------------------------------------------|
| Process investigated | Downwelling of coastal waters                        | Mild/shallow DSWC event                                                                                                                   |
| Period studied       | March 2011                                           | Winter-spring 2021-2022 (with detailed observations in March 2022)                                                                        |
| Instrumentation      | Moorings + CTD transect at the southern canyon flank | Multi-platform approach: moorings + hydrographic transects (glider + ship-based CTD) at the shelf and canyon + ADCP + reanalysis product. |

|                                |                                                                                                                                                                                    | Original observations and new data collected during the FARDWO-CCC1 Cruise (2022).                                                                                                                                                                               |
|--------------------------------|------------------------------------------------------------------------------------------------------------------------------------------------------------------------------------|------------------------------------------------------------------------------------------------------------------------------------------------------------------------------------------------------------------------------------------------------------------|
| Main forcings                  | Storm-induced downwelling                                                                                                                                                          | Moderate net heat losses during winter. Eastern storms further enhanced DSWC                                                                                                                                                                                     |
| Water mass characteristics     | Pot. temp.~ 11.5-12.5 °C
Salinity ~ 37.7
Pot. density ~ 28.78 kg·m -3                                                                                             | Pot. temp.~ 12.2-12.7 °C
Salinity ~ 38.1-38.2
Pot. density ~ 28.9-29.1 kg·m-3                                                                                                                                                                              |
| Dense water transports         | No significant dense shelf water formation.  Downwelling to the canyon head ~0.2 Sv and 10 5 t of SPM                                                                   | Dense shelf water formed in the Gulf of Lion during winter 2021-2022.  0.7 Sv and 10 5 t SPM cont. shelf 0.3 Sv and 10 5 t SPM upper canyon 0.05 Sv and 10 4 t SPM mid canyon                                                   |
| Detachment depths              | 200-300 m depth                                                                                                                                                                    | 150-400 m depth                                                                                                                                                                                                                                                  |
| Main contribution of the paper | Documentation of a storm-induced downwelling event, with the absence of DSWC under limited external forcing during a mild winter.  No comparison with previous events is provided. | Detailed observational characterization of a mild/shallow DSWC event, as well as the shelf-slope dense-water and sediment transport during a mild winter. Comparison with previous mild and extreme events (broader spectrum of cascading events in the region). |

We believe that all these elements demonstrate that our manuscript presents original observations of a shallow/mild cascading event, extending previous findings in the canyon under similar meteorological regimes, and contributing to a better understanding of the interannual variability of both mild and intense DSWC events in the Cap de Creus Canyon. We have made the pertinent changes throughout the manuscript to highlight these differences. We hope these revisions have addressed her concerns about length, focus, and balance. We hope these revisions have addressed your concerns about length, focus, and balance.

**Please find my comments below**

L20-23: I still find this sentence a bit ambiguous regarding the magnitude and extent of the DSW cascading and export. Mainly it is the term "export" what bothers me. What does "export" refer to? Can we use this term when we find the given water mass to be transported 100 km away? 10 km away? 1 km away?

**Reply:** We thank you for pointing this out. We agree that "export" can imply that dense shelf waters reach the open sea or deeper parts of the basin, which is not the case for the event studied here. To avoid ambiguity, we have replaced "export" with "transport" in the manuscript. Here, "transport" refers to the movement of dense shelf waters from the adjacent continental shelf into and along the canyon, reaching at least transect T2, approximately 30 km from the shelf. We have included this sentence into the abstract to clarify: "Dense shelf waters were transported ~30 km from the continental shelf into the canyon" (line 21).

L47: This is what I understand by "export".

**Reply:** Please, see previous comment.

**Methods.**

I appreciate the authors gave us more information about the interpolation (or gridding) method. But I still think more details are needed, mostly about the spacing of the grid or the length scales used (if so).

**Reply:** As stated before, the hydrographic profiles were interpolated using the isopycnic method integrated in the Data-Interpolating Variational Analysis (DIVA) software included in Ocean Data View (ODV) (v. 5.7.2). The interpolation parameters were set with scale lengths of 200 m horizontally and 1 m

vertically, a quality limit of 3.0, and excluding outliers. However, it is important to note that the griding primarily follows the isopycnals. Therefore, although the scale lengths influence the smoothing, the resulting gridded fields are mainly influenced by the isopycnal structure. The main advantage of this approach is that it improves the representation of water masses and reduces the artificial smoothing that the standard depth-based gridding method in ODV would do.

As suggested, we have better clarified this, as well as the spacing of the grid between lines 179-185.

L.234. SPM has not been defined at this point

**Reply:** Thank you for pointing this out. We have now defined it in the Abstract (line 15) and Introduction (line 113).

**Results. Section 4.1.**

I find a bit hard to interpret this section if I still don't know what, how and when, is related to the dense water export.

**Reply:** We agree that in the previous version we described the winter 2021-2022 conditions without sufficiently framing them in relation to dense water formation and the presence of dense shelf waters in the canyon. To address this, we have thoroughly revised Section 4.1 (lines 328-363) to link the meteorological and oceanographic forcings with the timing and presence of dense shelf waters. In particular, we provide context for both the period prior to the FARDWO-CCC1 cruise (highlighting the strong heat losses, persistent northerly and northwesterly winds that favored dense water formation in the GoL) and the subsequent period after the cruise with easterly storms, increased Hs, and enhanced coastal river discharges.

L361-363. Actually, the strongest ocean heat loss starts around November 1st and goes until approx. the end of February.

**Reply:** We have modified this in lines 333-335, which now reads: "The time series of net heat fluxes over the GoL's shelf showed negative values from October 2021 to early April 2022, indicating a heat loss from the ocean to the atmosphere (Fig. 3a). The strongest net heat losses during that winter occurred between November 2021 and late February 2022, reaching values of about -400 W·m $^{-2}$  (Fig. 3a).

L334-335. Why should discharge be related to the wind?

**Reply:** We have shortened this section to avoid excessive descriptive details and removed the previous sentence about the December peak discharge. However, the sustained easterly winds observed during March 11-13 favored the advection of Mediterranean humid air towards southern France, causing intense rainfall from the Eastern Pyrenees to the Massif Central, especially in the Aude and Hérault watersheds. In contrast, precipitation in the rest of France, including the Rhône River watershed, were weak (<a href="https://www.eaufrance.fr/publications/bsh/2022-04">https://www.eaufrance.fr/publications/bsh/2022-04</a>). As a consequence, coastal river discharges increased up to 2265 m3·s-1, while the Rhône River discharge remained comparatively lower (884 m3·s-1) (Fig. 3d)". You can find this explanation in lines 351-355.

L336-340. And related to the Hs, isn't it?

**Reply:** Indeed, the eastern storm on March 13 was associated with Hs > 3 m for over 20 hours. We have rephrased this paragraph to make this clearer (lines 351-355). According to the existing literature, easterly winds can produce large waves over the continental shelf, and lead to an intense cyclonic circulation on the GoL's shelf and to a strong export of shelf waters at the southwestern exit of the GoL (Ulses et al., 2008a; Mikolajczak et al., 2020).

Main comment: L397-398. I feel like this sentence should be the beginning of the story. The authors chose to provide the context before showing the presence of Dense Shelf Water, which is the object of the study. As I said in my previous comment, all that previous information is kind of empty if we don't

know where is the DSW observation and how it looks like. It is a matter of style maybe, but I'd find it much clearer the other way around.

**Reply:** We understand your point, and agree that in the previous version the link between dense water formation and the presence of dense shelf waters (DSW) in the canyon was not fully framed. We have chosen to maintain the structure presenting the meteorological, oceanographic, and hydrological context of winter 2021-2022 before showing the DSW observations in the canyon, as this allows to have a clearer understanding of the background conditions. We agree that in the previous version, it was difficult to locate the observations in space and time. With the new revisions on sections 4.1 and 4.2, we believe the context now sufficiently frames our observations.

Fig 4e. Why is the y-axis scale so large? You could reduce them a lot so the variability would be much more visible. Currents seem to be nearly zero at the CCC during the cruise time, which is actually the focus of this study. I also wonder what does the alternating positive negative current pattern means. Do you have an explanation for the up-canyon and down-canyon currents to be so regular?

**Reply:** Thank you for your comments.

Fig. 4e: We have reduced the y-axis range to 1 mg·L-1 to better illustrate the temporal variability of SPM concentrations.

Current speeds during the FARDWO-CCC1 cruise: Indeed, current speeds at 1000 m depth in the Cap de Creus Canyon were relatively low during the cruise. This indicates that dense shelf waters did not reach the bottom at the mooring location during the winter 2021-2022. Our observations are consistent with a mild cascading event reaching depths of ~400 m depth in the canyon.

Alternating positive-negative current patterns in LDC-1000: Previous studies (e.g., Béthoux et al., 2002) have reported near-bottom oscillatory currents of 0.1 m·s-1 at 1000 m depth in the Lacaze-Duthiers Canyon associated with intense dense shelf water cascading events. However, during our study, the temperature time series in the canyon and the atmospheric and oceanographic conditions during winter 2021-2022 do not show any evidence of deep cascading, which is usually associated with these oscillations at these depths. Likely, these oscillations could be related to the interaction of the meandering Northern Current (offshore displacements) with canyon topography, as previously interpreted by Durrieu de Madron et al. (1999) for the Grand-Rhône Canyon.

Fig 5. Please make the axes labels larger.

**Reply:** Done. We have done it for figures 5, 6, and 7.

L406. I can't see the high dissolved oxygen as compared with the upper layer.

**Reply:** We have corrected the text to indicate that dissolved oxygen values in this layer were around 200  $\mu$ mol·kg-1 (line 417).

L461-463. Try to avoid subjective language, mostly when numbers are provided, and let the reader decide. 150 W m-2 is 25% less than 200 W m-2, and is half of 500 W m-2.

**Reply:** Thank you for pointing this out. We have rephrased the sentence to express the differences quantitatively (lines 508-510).

Discussion: I find the discussion to be way too long. While the writing is very clear and the connection with the bibliography is excellent, the new discussion is substantially longer than in the previous version, with many additions for which I do not really see the relevance (unless it's in response of the reviewer comments, but even then..). There are also lots of repetitions of the results numbers and references to figures. I would recommend to summarize and streamline, I've made more concrete suggestions about this in the following comments.

**Reply:** We appreciate your comment and agree with your suggestions. Some of the additions in the previous version were introduced in response to earlier comments from the reviewers. However, after rewriting the Discussion, we realized that some parts did not connect well with the text and added little value. Therefore, we have summarized and streamlined the Discussion to make the message clearer and specially to avoid repetitions with the Results section. At the same time, we have reorganized it to improve its logical progression and included some more discussion around certain topics (as for Fig. 10) to highlight the results. Please, find the modified Discussion from line 504.

L495-496. But as important as talking about current speed is current direction.

**Reply:** We have removed the detailed description of the mooring data to avoid repetitions with the Results section. Also, we agree that both current speed and direction are important. The mooring data indicated that there was no significant down-canyon flow at the monitored depths by the moorings, confirming the absence of a deep cascading during the studied winter.

L519-535. I cannot see the point of these lines. This part seems disconnected from the rest of the study, it does not really provide any useful information, and is also misplaced in the discussion section.

**Reply:** We understand your concerns and agree that the paragraph describing the hydrodynamics of the plume using the Richardson (Ri) and Froude (Fr) numbers may seem disconnected from the main focus of the manuscript. This section was originally included in response to a previous suggestion from the other reviewer to examine the hydrodynamics of the dense water plume. However, after a careful consideration, we agree that the detailed discussion of Ri and Fr does not directly contribute to the discussion of the MDSWC event. Therefore, we have removed this paragraph from the revised manuscript. Nonetheless, we acknowledge that this information is of high interest for a separate study specifically focused on the physical dynamics of dense shelf water cascading.

L551-569. I find this paragraph too long. Mostly because it only addresses a small part of the results, the SPM concentrations in the plume, which is also not so surprising.

**Reply:** Thank you for your comment. However, we believe that this section actually provides valuable information, since one of the main goals of the paper is to investigate SPM transports associated with cascading. We acknowledge that in the previous version these results were not sufficiently discussed, but we have now placed them in a new section (5.3) and expanded the discussion. We now emphasize how the observed SPM values in March 2022 could be comparable to those expected for the entire cascading season identified in the reanalysis, although we acknowledge that they are estimates.

L582-583. This is an example of results repetition than can be avoided.

**Reply:** We agree with your comment. To address this, we first have rephrased the sentence as: "The transports of dense shelf waters and associated SPM in the Cap de Creus Canyon during the observed MDSWC event in March 2022 were 0.7 Sv and 105 t across the continental shelf, 0.3 Sv and 105 t in the upper-canyon, and 0.05 Sv and 104 t in the mid-canyon". We have also moved this to a new section in Results, entitled "4.4. Duration and magnitude of cascading events during winter 2021-2022" (lines 463-503).

L591- From here, I'd say this belong to results and not to the discussion section. Also, the comparison would be more clearer with a map (even in the S.I), as with the data provided in the table we can't really see how well the spatial patterns are represented. For instance, the reanalysis provides 12.42°C for T1-03, and T104, while their observed mean temperature differed by 0.4°C.

**Reply:** We agree with your comment and have incorporated this information to new section 4.4 (lines 463-503). Additionally, to clarify the spatial comparison between observations and reanalysis, we have included a new figure in the Supplementary information (new Fig. A1) illustrating the spatial distribution of stations. As you pointed out, there are some apparent differences in temperature values, which mainly arise from the different spatial resolution of the MedSea reanalysis (~4.5 km). The T1 and T2 observational transects did not follow a perfectly straight line, and stations were spaced on average 1.5 km. In contrast,

reanalysis stations are located in idealized straight transects, with each point corresponding to the nearest model grid cell. As a result, two nearby reanalysis stations may share the same temperature value because they fall within a single grid cell, whereas nearby observational stations can display different temperatures. Inevitably, reanalysis slightly smooths small-scale variability, especially in narrow submarine canyons, but the overall agreement between the two datasets is still high, with RMSE < 0.2 °C.

L605-606 I guess this means that the transport has been averaged for the dense waters density layers. But then, There are many instances with up-canyon transport, which is quite weird, isn't it? Or is there an explanation for this?

**Reply:** Thank you for your comment. Transport values were calculated by integrating the along-canyon velocity over the dense water density layer. Occasional up-canyon transports during the cascading season likely reflect short-term current reversals along the canyon axis. However, when we integrate transport over time, the net flux is down canyon during the cascading season, consistent with the direction of dense water overflows.

After the cascading season, the transport shifted predominantly up-canyon (data not shown), which reflects the residual flux along the canyon axis. This residual flux has been previously documented within the Palamós Canyon, where a persistent up-canyon flow is superimposed on the periodic (i.e., inertial) along-canyon oscillations (Martín et al., 2006, 2007; Arjona-Camas et al., 2021). Please, see lines 489-495.

Figure 10 is nice, but I miss more discussion around it. Also, can you explain how did you differentiated between mild and intense events? Is this a wind, or heat loss threshold as you mentioned in the introduction? It is surprising that transport is often as intense in mild events as in intense events.

**Reply:** In figure 10 we aimed to provide a long-term context of the interannual variability of dense shelf water cascading events through the Cap de Creus Canyon. As explained in the Methodology (Section 3.5), we distinguished between MDSWC and IDSWC events based on density thresholds: IDSWC were defined by densities > 29.1 kg·m-3 and MDSWC events with densities below this threshold. Regarding the forcing mechanisms, the distinction of MDSC and IDSWC events is not solely based on wind or heat loss thresholds, but a combination of them. Nevertheless, the exact triggering mechanisms and magnitudes driving DSWC need further analysis (Fos et al., 2025).

Indeed, it is surprising that the magnitude of transport during mild winters is sometimes comparable to that of intense winters. As previously acknowledged in the literature (Mikolajczak et al., 2020) and discussed in this paper, the major difference between mild and intense cascading events may concern more the preferential transport pathways than the volume exported. In winter 2010-2011, only 30% of the transport occurred through the canyon, while 70% followed along the coast or remained around the upper canyon. In winter 2004-2005, 69% of dense shelf water cascaded through the Cap de Creus Canyon down to the deeper parts of the canyon (Ulses et al., 2008a; Mikolajczak et al., 2020).

**References:**

Arjona-Camas, M., Puig, P., Palanques, A., Durán, R., White, M., Paradis, S., and Emelianov, M.: Natural vs. trawling-induced water turbidity and suspended sediment transport variability within the Palamós Canyon (NW Mediterranean). Mar. Geophys. Res. 42, 38, https://doi.org/10.1007/s11001-021-09457-7, 2021.

Béthoux, J., Durrieu de Madron, X., Nyffeler, F., and Taiiliez, D.: Deep water in the western Mediterranean: peculiar 1999 and 2000 characteristics, shelf formation hypothesis, variability since 1970 and geochemical interferences. J. Mar. Sys. 33, 117-131, http://doi.org/ 10.1016/S0924-7963(02)00055-6, 2002.

Durrieu de Madron, X., Radakovitch, O., Heussner, S., Loye-Pilot, M. D., and Monaco, A.: Role of the climatological and current variability on shelf-slope exchanges of particulate matter: Evidence from

the Rhône continental margin (NW Mediterranean). Deep Sea Res. I Oceanogr. Pap. 46, 1513, https://doi.org/10.1016/S0967-0637(99)00015-1, 1999.

Fos, H., Izquierdo-Peña, J., Amblas, D., Arjona-Camas, M., Romero, L., Estella-Pérez, V., Florindo-Lopez, C., Calafat, A., Cerdà-Domènech, M., Puig, P., Durrieu de Madron, X., and Sanchez-Vidal, A.: Solving dense shelf water cascading with a high-resolution ocean reanalysis. ESS Open Archive, March 17, https://doi.org/10.22541/essoar.174060515.57729804/v2, 2025.

Martín, J., Palanques, A., and Puig, P.: Composition and variability of downward particulate matter in the Palamós submarine canyon (NW Mediterranean). J. Mar. Sys. 60, 75-97, https://doi.org/10.1016/j.jmarsys.2005.09.010, 2006.

Martín, J., Palanques, A., Puig, P.: Near-bottom horizontal transfer of particulate matter in the Palamós submarine Canyon (NW Mediterranean). J. Mar. Res. 65, 193-218, https://doi.org/10.1357/002224007780882569, 2007.

Martín, J., Durrieu de Madron, X., Puig, P., Bourrin, F., Palanques, A., Houpert, L., Higueras, M., Sánchez-Vidal, A., Calafat, A. M., Canals, M., Heussner, S., Delsaut, N., and Sotin, C.: Sediment transport along the Cap de Creus canyon flank during a mild, wet winter. Biogeosciences 10(5):3221-3239, https://doi.org/10.5194/bg-10-3221-2013, 2013.

Mikolajczak, G., Estournel, C., Ulses, C., Marsaleix, P., Bourrin, F., Martín, J., Pairaud, I., Puig, P., Leredde, Y., Many, G., Seyfried, L., and Durrieu de Madron, X.: Impact of storms on residence times and export of coastal waters during a mild autumn/winter period in the Gulf of Lion. Cont. Shelf Res. 207, 104192, https://doi.org/10.1016/j.csr.2020.104192, 2020.

Ulses, C., Estournel, C., Bonnin, J., Durrieu de Madron, X., and Marsaleix, P.: Impact of storms and dense water cascading on shelf-slope exchanges in the Gulf of Lion (NW Mediterranean). J. Geophys. Res. Oceans, 113, http://doi.org/10.1029/2006JC003795, 2008a.

**Response to reviewer 2 (2nd revision)**

**Dear reviewer,**

We thank you very much for your constructive and relevant comments to our manuscript. Below, your reviews are reproduced in **black** font and our responses in **blue**. Since the other reviewer has raised important points, we kindly suggest to review her responses. Please, note that all line numbers in our responses refer to the **clean version** of the manuscript, not the tracked-changes version.

**Review Comments:**

The authors have substantially improved the manuscript in response to the previous round of comments. The study presents a well-structured observational analysis of dense shelf water cascading (DSWC) and sediment transport in the Cap de Creus Canyon during a mild winter regime. The multi-platform dataset (moorings, gliders, CTD profiles, and reanalysis products) is robust, and the results provide useful insights into the dynamics, timing, and sediment export processes under mild winter conditions.

**Reply:** We thank you very much for your encouraging comment.

**Novelty Assessment:**

The novelty is somewhat limited because DSWC in the Cap de Creus Canyon under mild winter conditions has previously been described by *Martín et al.* (2013) for the 2010–2011 winter, including estimates of dense water transport ( $^{\circ}0.3$  Sv) and sediment load ( $^{\circ}10^{5}$  t). The present study adds:

- A more recent mild-winter case (2021–2022) with higher-resolution, multiplatform observations.
- Measurements across both the continental shelf and canyon transects.
- Integration of hydrodynamic and sediment transport data with updated reanalysis products.

At present, the novelty is primarily methodological and contextual rather than conceptual. However, it can be strengthened by including an explicit quantitative comparison of transport and sediment load values between:

- 1. The present mild winter (2021–2022),
- 2. The previous mild winter (2010–2011; Martin et al., 2013), and
- 3. Known strong-winter events (e.g., Canals et al., 2006; Puig et al., 2008).

Such a comparison would position the study as the first to place recent mild-winter dynamics into the broader spectrum of DSWC intensities in the Cap de Creus Canyon, increasing its interpretive value and relevance for understanding climate driven variability in cascading processes.

**Reply:** Thank you for your suggestion. In line with yours and the other reviewer's comment, we first have streamlined the Discussion to avoid repetitions with the Results section, and then expanded the Discussion on certain paragraphs, such as around Fig. 10. We have now included a direct comparison between winter 2021-2022 with both previous mild and strong winters. This comparison considers the atmospheric forcings and transport values, and allows us to put our study within the broader spectrum of cascading intensities in the Cap de Creus Canyon. Please, see the new Discussion from line 505.

**Abstract Clarity:**

The sentence "...yet its dynamics under mild winter regimes remain poorly characterized" should be qualified to avoid implying a global knowledge gap. Since mild-winter DSWC has been documented elsewhere (e.g., Mahjabin et al., 2019, 2020), and even in the Cap de Creus Canyon (Martin et al., 2013), I recommend revising to this line. For example it can be written as:

"...yet its dynamics under mild winter regimes in the northwestern Mediterranean, particularly in the Cap de Creus Canyon, have been less comprehensively described and compared to strong-winter events."

This way:

- It narrows the scope to **region + site** (avoids implying a global knowledge gap).
- It acknowledges some existing work (e.g., Martin et al. 2013) but still justifies the new study.
- It sets up the importance of comparison with strong winters early in the paper.

**Reply:** Thank you for your suggestion. We have revised the sentence to narrow the scope, which now reads: "Although intense DSWC events have received most attention due to their large impacts, mild DSWC (MDSWC) events are the most frequent in the northwestern Mediterranean and are expected to become more common under climate change. However, their dynamics, particularly in the Cap de Creus Canyon, have been less comprehensively described and compared to strong-winter events".

We have also added a new sentence at the end of the abstract to highlight the variability of DSW transports in the Cap de Creus Canyon, even under mild winters, which reads: "Our study reinforces the idea that dense shelf water transports exhibit marked interannual variability, even under mild winters" (lines 26-27).

**Minor Corrections and Consistency Edits:**

• SI unit for metric tonnes – Use the correct SI symbol: t (lowercase). At first occurrence, write as t (metric tonnes), and thereafter use t alone. Ensure a space between the value and the unit (e.g., "105 t", not "105t"). Replace non-SI or ambiguous forms such as "metric tons" or "T" where applicable.

**Reply:** Done. We have changed it throughout the manuscript and ensured that there is a space between the value and the unit as recommended.

• **Hyphenation** – Standardize usage to either *dense shelf water cascading* (no hyphen) or *dense shelf-water cascading* (with hyphen) throughout text and captions.

**Reply:** We have standardized it throughout the manuscript and captions, using *dense shelf water cascading* (without hyphen).

• **Acronyms** – In Section 3.2.1, correct ECMWF to *European Centre for Medium range Weather Forecasts* (ECMWF).

Reply: Done.

**Overall Recommendation:**

With these relatively minor edits and an expanded discussion comparing the present results with both previous mild-winter and strong-winter events, the manuscript will be well-prepared for publication in Ocean Science. The observational dataset is valuable, the analyses are sound, and the study adds meaningful insight into DSWC dynamics in a mild winter regime for this specific canyon system.

Reply: We thank you very much for your overall recommendation. We have re-structured and streamlined the Discussion (as suggested by the other reviewer) and expanded our discussion in order to compare our results with both previous mild and strong-winter events. We believe that with these changes, we have strengthened our understanding in shelf-slope exchanges during MDSWC events in the Cap de Creus Canyon, their interannual variability, and their relevance under climate change scenario.

---

## Author Response (AR3)

**Response to reviewer 1**

Dear reviewer,

We thank you very much for your overall comments to our manuscript. Below, your reviews are reproduced in black font and our responses in blue.

**Review Comments**

The authors have substantially improved the manuscript in response to previous reviewer comments. The study presents a solid observational analysis of dense shelf water cascading (DSWC) and sediment transport in the Cap de Creus Canyon during the mild winter of 2021–2022. The dataset is comprehensive (moorings, gliders, CTD profiles, and reanalysis products), and the comparisons with both previous mild-winter and strong-winter events now place the results in the correct broader context. The manuscript is much clearer and better framed than earlier versions.

**Remaining Issues**

**1. Table 1**

Table 1 is useful for transparency (particularly for RMSE validation values and comparison with *Martín et al., 2013*), but the essential findings (RMSE consistently <0.2 °C; quantitative comparability across events) are already described in the text. To streamline the manuscript, I recommend moving Table 1 to the Supplementary Material and retaining only a summary statement in the main text.

**Reply:** Thank you for your suggestion. We have moved Table 1 to Appendix 1 and renamed it as Table A1. We have also updated the appendix title to "Comparison between in-situ observations and reanalysis data across the Cap de Creus Canyon" to better reflect its content.

**2. Minor corrections**

• **Acronyms** – In Section 3.2.1, correct ECMWF to *European Centre for Medium-range Weather Forecasts* (ECMWF).

**Reply:** Changed.

• **Hyphenation** – Standardise usage of *dense shelf water cascading* (no hyphen) or *dense shelf-water cascading* (with hyphen) consistently across text and figure captions. So to be consistent just remove the hyphen from Title.

Reply: Changed.

**Overall Recommendation**

With these final minor corrections, the manuscript will be well-prepared for publication in *Ocean Science*. The observational dataset is valuable, the analyses are sound, and the paper now provides meaningful and well-contextualized insights into DSWC dynamics in a mild winter regime for the Cap de Creus Canyon.

**Reply:** We are glad that the revisions have improved the message and overall quality of the manuscript. We appreciate the positive feedback and the final recommendation for publication.